

# Assessing BERT-based models for Arabic and low-resource languages in crime text classification

Njood K. Al-harbi and Manal Alghieth

Department of Information Technology, College of Computer, Qassim University, Buraydah, Saudi Arabia

## ABSTRACT

The bidirectional encoder representations from Transformers (BERT) has recently attracted considerable attention from researchers and practitioners, demonstrating notable effectiveness in various natural language processing (NLP) tasks, including text classification. This efficacy can be attributed to its unique architectural features, particularly its ability to process text using both left and right context, having been pre-trained on extensive datasets. In the context of the criminal domain, the classification of data is a crucial activity, and Transformers are increasingly recognized for their potential to support law enforcement efforts. BERT has been released in English and Chinese, as well as a multilingual version that accommodates over 100 languages. However, there is a pressing need to analyze the availability and performance of BERT in Arabic and other low-resource languages. This study primarily focuses on analyzing BERT-based models tailored for the Arabic language; however, due to the limited number of existing studies in this area, the research extends to include other low-resource languages. The study evaluates these models' performance in comparison to machine learning (ML), deep learning (DL), and other Transformer models. Furthermore, it assesses the availability of relevant data and examines the effectiveness of BERT-based models in low-resource linguistic contexts. The study concludes with recommendations for future research directions, supported by empirical statistical evidence.

## INTRODUCTION

Criminality is a detrimental phenomenon that impacts both developed and underdeveloped nations globally. Criminal actions can significantly disrupt the nation's economy and negatively impact people's lives, resulting in societal issues (*Safat, Asghar & Gillani, 2021*). Criminal detection and prediction constitute an essential practice in crime analysis, seeking to provide effective techniques for supporting law enforcement (*Tam & Tanrıöver, 2023*). Researchers develop diverse strategies and solutions utilizing machine learning (ML) and deep learning (DL) algorithms to foresee or detect criminal activities that have demonstrated considerable potential in addressing crime detection issues (*Mandalapu et al., 2023*).

Corresponding author
Njood K. Al-harbi,
441212541@qu.edu.sa

In addition, Transformer are recent DL paradigms that have gained attention in various research domains, including natural language processing (NLP), computer vision (CV), and speech processing (*Petrolini, Cagnoni & Mordonini, 2022*). Recent studies demonstrated that Transformer-based pre-trained models (PTMs) can achieve state-of-the-art performance across several tasks, particularly in NLP (*Qiu et al., 2020*). NLP is a technique that can be used to examine unstructured data sources, including news articles, social media posts, and police reports, to extract essential information regarding criminal activity.

Text classification is a crucial task in the field of NLP across several applications such as topic classification, question answering, and sentiment analysis which is the most popular use (*Kora & Mohammed, 2023*), covering binary and multiple categories. Binary classification refers to categorizing data into two distinct categories, while multi-classification involves classifying data into more than two categories.

Now let us shift our focus to the very new Transformer, BERT is a Transformer model that was widely used last year for text classification tasks (*Liu, Wang & Ren, 2021*). The BERT framework was introduced through two steps: pre-training and fine-tuning. Initially, during pre-training, the model was trained on unlabeled data across various tasks. Subsequently, in the fine-tuning step, the BERT model was initialized with the pre-trained parameters, and it was then refined using labeled data for downstream tasks. Figure 1 shows Comprehensive BERT pre-training and fine-tuning.

It follows a self-supervised learning approach and consists of two main functions: the masked language model (MLM) and the next sentence prediction (NSP) (*Devlin et al., 2018*; *Devlin, 2018*).

MLM is the process of randomly masking some words in input data. The goal is to estimate the original vocabulary of the masked words by considering the context provided by the unmasked words. MLM reads the sentence bidirectionally, both from left to right and from right to left, allowing the model to gain a comprehensive understanding of the language context. That's contrasted to other pre-trained language models that only read unidirectionally from left to right or from right to left (*Devlin et al., 2018*). In addition to MLM, NSP is used to pre-train BERT. NSP involves feeding the model two sentences and asking it to determine whether the second sentence in the pair is the subsequent sentence to the first. This method facilitates the comprehension of the relationships between sentences, and it is a crucial capability in NLP tasks such as summarization, question-answering, and text classification (*Devlin et al., 2018*).

The original BERT article introduces two variations: BERT large and BERT base. Each variation supports both cased and uncased input text. The training process exclusively utilizes raw English text without any human involvement in the labeling process (*Devlin, 2018*). However, there exist several versions of BERT that have been developed to support different languages such as multilingual BERT (mBERT) (*Devlin, 2018*), XLM-RoBERTa (XLM-R) (*Conneau et al., 2019*), and DistilBERT (*Sanh et al., 2019*). Some of these versions are specifically designed for distinct languages, such as AraBERT (*Antoun, Baly & Hajj, 2020*), MARBERT, ARBERT (*Abdul-Mageed, Elmadany & Nagoudi, 2020*), and ArabicBERT for Arabic, bert-base-chinese for Chinese (*Devlin, 2018*), AM-BERT,
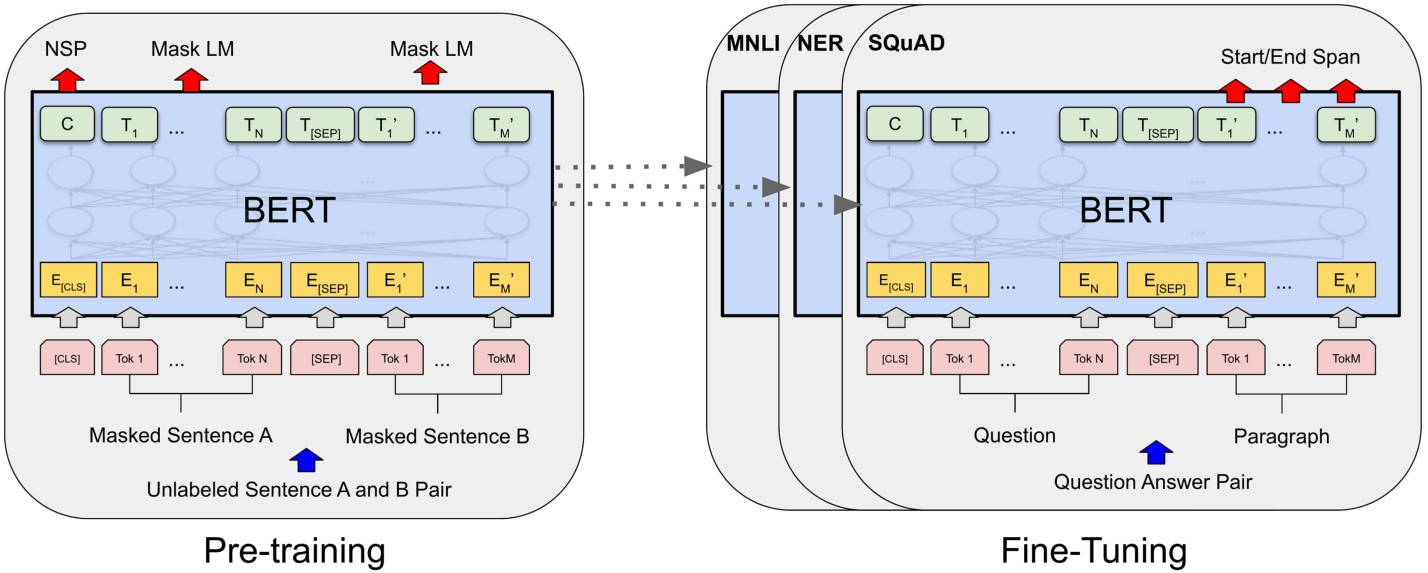

**Figure 1 Comprehensive BERT pre-training and fine-tuning.**

AM-RoBERTa for Amharic (*Yimam et al., 2021*), FlauBERT for French (*Le et al., 2020*), and BanglaBERT for Bangla (*Bhattacharjee et al., 2021*). Furthermore, there are specialized versions of BERT specialized to certain domains, such as ClinicalBERT for the clinical domain (*Alsentzer et al., 2019*), LEGAL-BERT for the legal domain (*Chalkidis et al., 2020*), and FinBERT for the financial domain (*Araci, 2019*).

Previous research on ML and DL has shown effective results in identifying Arabic criminal textual data using models such as eXtreme Gradient Boosting (XGBoost) and SHapley Additive exPlanations (SHAP), as indicated by *Taiwo, Saraee & Fatai (2024)* and using CNN by *Abdalrdha, Al-Bakry & Farhan (2023)*. Additionally, the article by *Monika & Bhat (2022)* achieved commendable results by employing a hybrid wavelet convolutional neural network with World Cup Organization (WCNN-WCO). Moreover, the study by *Hissah & Al-Dossari (2018)* investigates multiple ML approaches in detecting crimes, including support vector machines (SVM), decision trees (DT), complement naive Bayes (CNB), and k-nearest neighbors (KNN). demonstrated a good performance.

Lastly, in another domain that endorsed the utilization of BERT models, the article by *Bahurmuz et al. (2022)* achieved commendable results in rumor identification utilizing transformer-based BERT. Additionally, various other fields, such as sentiment analysis, have validated the capability of BERT base models in detecting criminal text data for Arabic languages such as *Chouikhi, Chniter & Jarray (2021)*, *Althobaiti (2022)*, and *Abdelgwad, Soliman & Taloba (2022)*. The purpose of focusing on BERT is its capacity to support numerous languages and its various pre-trained models, which should be evaluated for effectiveness in the specific field of crime.

This study will discuss different BERT models, including low-resource and high-resource languages to classify criminal data from various sources. It aims to clarify

the availability of research within the crime domain and the dataset's availability, as well as its impact on performance compared to previous ML, DL, and Transformers models.

The study first concentrates on Arabic as a primary language, but due to the existence of only one research finding, it has been broadened to encompass multiple languages. Consequently, the current study contributes a deep analysis of studies in low-resource and high-resource languages that employ a BERT-based model for the classification of criminal text. The main contributions of the current review are:

- Provide statistics and facts from studies in the field of crime using transformers.
- This study provides a full analysis of the transformer model architecture employed in every study work.
- Specify the classification types and different categories of crimes.
- Fully evaluation of the dataset's availability, consistency, and completeness of the rules.
- Evaluate the metrics employed throughout the implementation.
- Investigate the comparison of ML, DL, and Transformers, alongside previous studies.
- Examine the best findings and the worst findings, together with the underlying reasons for both findings.
- Assess the models that accommodate several languages as well as those specialized for a single language.
- The primary distinctions between studies on high-resource and low-resource languages.
- Shed light on new recommendations and enhancements for novel research on low-resource languages.

## RESEARCH METHODOLOGY

This study was conducted on the guidelines outlined in the Preferred Reporting Items for Systematic Reviews and Meta-Analyses (PRISMA) (*Moher et al., 2009*; *Page et al., 2021*) . The following subsections explain it in detail, and Fig. 2 summarizes them.

### Defying research question

This study aims to investigate the BERT model in classifying crimes focusing initially on Arabic text and then on texts in other languages. To achieve the aim using PICO helps to construct the question, which refers to Population, Intervention, Comparison or Control, and Outcome, which has been suggested by *Kitchenham & Charters (2007)*. The PICO strategy details in the current systematic review are as follows:

- Population: crime-related text (Tweets, General reports) in Arabic and other languages
- Intervention: BERT-based models
- Comparison: Compare BERT performance with traditional ML, DL, or Transformers models
- Outcome: classification accuracy, F1-score, precision, recall, and statistics

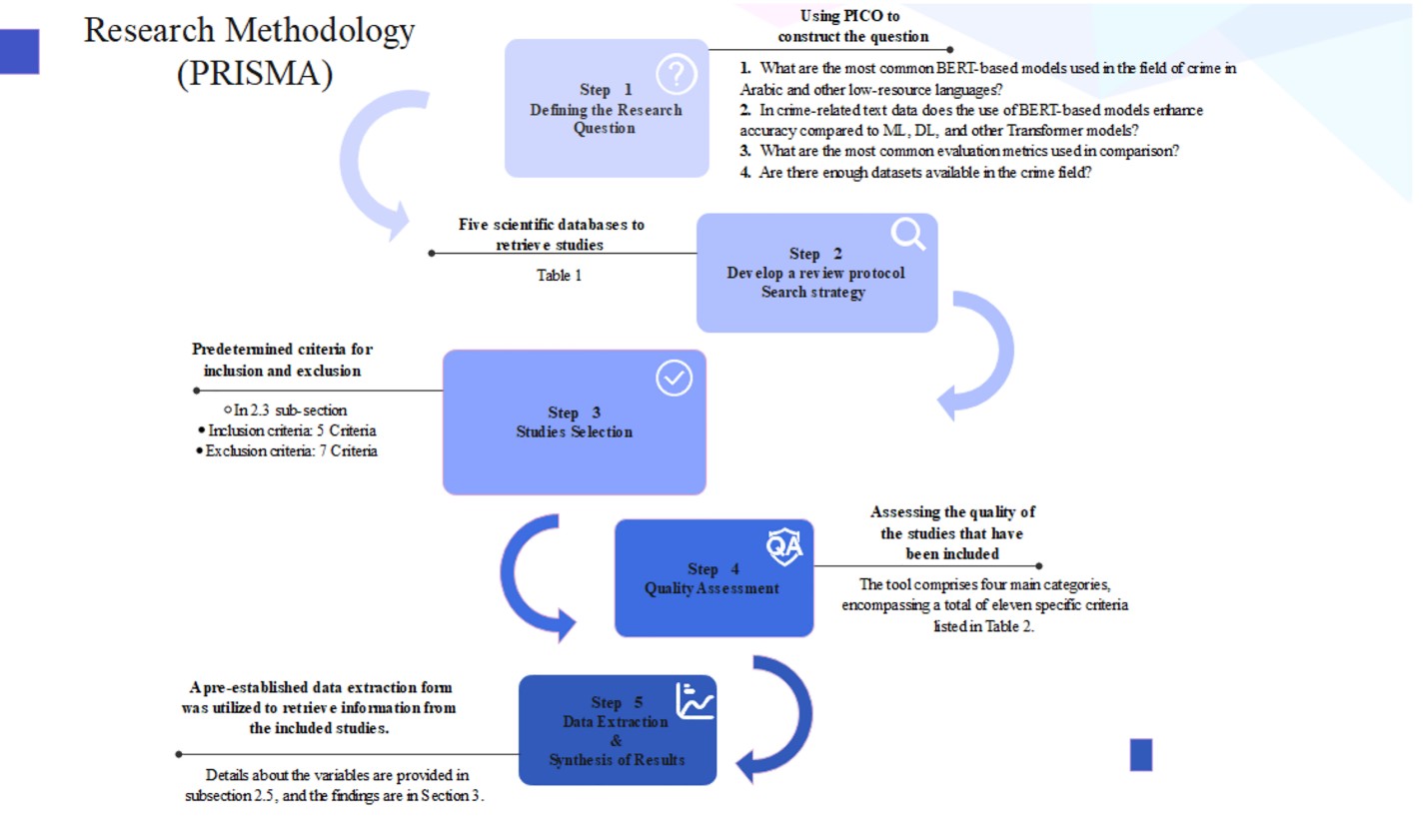

**Figure 2 Research methodology steps.**

The following are questions to achieve the objectives of the review:

- What are the most common BERT-based models used in the field of crime in Arabic and other low-resource languages?
- In crime-related text data, does the use of BERT-based models enhance accuracy compared to ML, DL, and other Transformer models?
- What are the most common evaluation metrics used in comparison?
- Is there enough dataset available in the crime field?

### Develop a review protocol search strategy

The current study relied on different scientific databases to retrieve studies: IEEE Xplore, Web of Science, ACM Digital Library, Taylor & Francis, and Google Scholar. These databases were considered for their cover and application in the sphere of technology. The date range is restricted to the period between 2018 and 2025. The year 2018 is significant since it marks the first release of BERT, while the latest search date is May 2025.

The search phrase was formulated by utilizing the PICO strategy again. This tool is utilized to design literature search strategies, particularly in the context of systematic reviews. Table 1 illustrates the search string employed for each database.

**Table 1 Search string in the databases.**

| Database name | Search string |
|---|---|
| IEEE Xplore | ("All Metadata":Bert OR "AIl Metadata":Bidirectional Encoder Representations from Transformers OR "All Metadata":"Arabic Bert") AND ("All Metadata":crime detection) AND ("All Metadata":text classification) |
| Web of Science | All=(("Arabic Bert" OR "Bert" OR "Bidirectional Encoder Representations from Transformers") AND ("crime detection" OR "text classification")) |
| ACM Digital Library | [All: bidirectional encoder representations from transformers] AND [All: bert] AND [All: Arabic bert] AND [All: crime detection] AND [All: crime text classification] AND [E-Publication Date: Past 5 years] |
| Taylor & Francis | [[All: bert] OR [All: Arabic]] AND [All: crime] AND [All: classification] AND [All: text] AND [All: bidirectional encoder representations from transformers] AND [Publication Date: (01/01/2018 TO 12/31/2025)] |
| Google Scholar | [bidirectional encoder representations from transformers AND bert AND Arabic bertAND crime detection AND crime text classification] |

## Studies selection

Once the studies were retrieved from the databases, they were imported into the reference management system, specifically Mendeley. Subsequently, any duplicate studies were eliminated. Then, the studies were screened by examining their titles and abstracts based on predetermined criteria for inclusion and exclusion, which will be explained below:

### Inclusion criteria

- Utilize BERT models for classification purposes rather than as an embedding strategy.
- Evaluation of the effectiveness of the model.
- The dataset is well described.
- The article is peer-reviewed (Journal articles or Proceedings articles).
- The article is written in the English language.

### Exclusion criteria

- The full text is not available online
- The articles are presented in all kinds of formats of gray literature (non-peer-reviewed) such as reports, posters, tutorials, presentations, theses, or dissertations.
- The article is not in English.
- Does not pertain to criminal activity
- Does not employ BERT as a primary model for classification
- Absence to assess the performance of the model.
- The dataset has not been described.

## Quality assessment

A tool derived from *Alammary (2022)* and *Zhou et al. (2015)* is utilized for assessing the quality of the studies that have been included. The tool consists of four primary criteria, which include a total of eleven criteria listed in Table 2 (*Dybå & Dingsøyr, 2008*). These

| Table 2 Assessment tool. | |
|---|---|
| **Four main criteria** | **The eleven criteria** |
| Reporting | 1. Does the article rely on empirical research or is it solely a report based on expert opinion? |
| | 2. Does the research have a well-defined set of aims? |
| | 3. Does the research provide a sufficient explanation of the context in which it was conducted? |
| Rigor | 4. Did the research design effectively address the objectives of the study? |
| | 5. Did the recruitment strategy align with the objectives of the research? |
| | 6. Was there a control group available for comparing treatments? |
| | 7. Was the data acquired in a manner that effectively addressed the research issue? |
| | 8. Was the data analysis conducted with enough rigor? |
| Credibility | 9. Has the extent to which the link between the researcher and participants has been thoroughly evaluated? |
| | 10. Does the report contain an unambiguous presentation of the research findings? |
| Relevance | 11. Is the study of value more beneficial for research or practice? |

main criteria include reporting, rigor, credibility, and relevance. The following criteria will be explained in depth:

- Reporting refers to the level of quality in the presentation of study objectives, rationale, and contextual information. Three criteria were related to reporting (1–3).
- Rigor refers to the comprehensive explanation of the research methodology used and the reliability of the dataset, including the tools used for data collecting and the procedures employed for data analysis. Five criteria were related to reporting (4–8).
- Credibility refers to the reliability and trustworthiness of the findings of the study. It is determined by the extent to which the data supports the findings and how effectively they are presented. Two criteria were related to reporting (9–10).
- Relevance refers to the extent to which a study's contribution is pertinent and significant to the domain or research community. One criterion was related to reporting (11).

As stated in *Zhou et al. (2015)*, the majority of researchers employed weights and a three-point scale to assess the extent to which the creation was met. In this scale, a score of 1 indicates fully met, a score of 0.5 indicates partially met and a score of 0 indicates not met (*Carrera-Rivera et al., 2022*; *Kotti, Galanopoulou & Spinellis, 2023*; *Semasaba et al., 2020*). Some researchers utilize a binary scale, assigning a value of 1 for fully met and 0 for not met (*Dybå & Dingsøyr, 2008*). However, in certain special cases, researchers may assign alternative weights to emphasize specific criteria within their system (*Alammary, 2022*). Researchers in *Zhou et al. (2015)* have determined that the binary scale is weaker than the three-point scale because it offers a restricted evaluation of quality details within the range between 0 and 1.

Thus, this systematic review will use the three-point scale. The sum of all the responses to each question will yield a quantitative measure (ranging from 0 to 11) that reflects the quality of the empirical study.

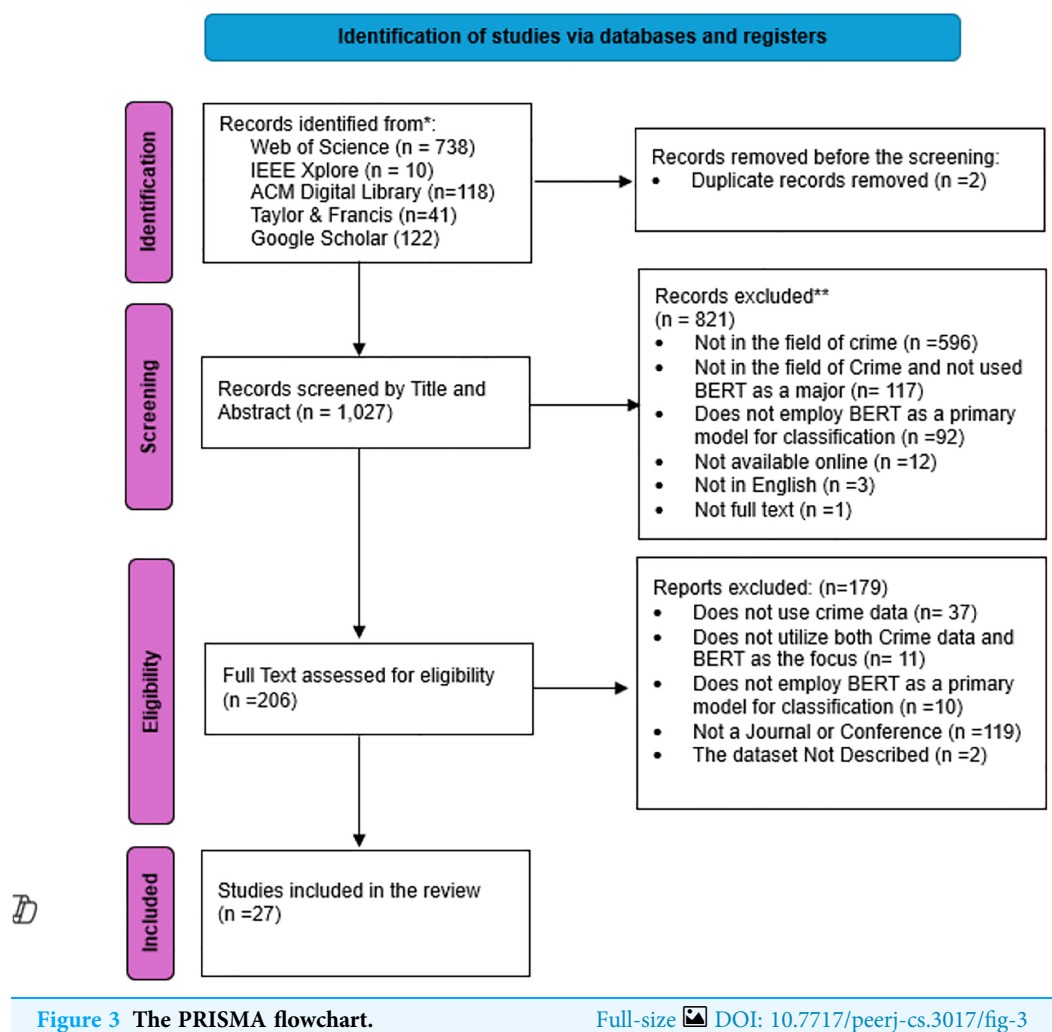

**Figure 3 The PRISMA flowchart.**     

## Data extraction and synthesis of results

Once the studies had been selected, a pre-established data extraction form was utilized to retrieve information from the included studies. The form contained the subsequent variables: The information includes the type of publication, the full name of the author, the title of the article, the title of the source, the language, the type of document, the title and location of the conference (if available), the keywords, the abstract, the number of cited references, the publisher, the city where the publisher is located, the year of publication, and the research areas.

Subsequently, a comprehensive examination of each study will be conducted to extract the contribution, classification type, dataset description, model utilized, along with its hyperparameters, and evaluation criteria. Upon extracting the data from the studies included, the results are synthesized to address the research questions. The information that has been extracted will be detailed in Section "Finding of SLR".
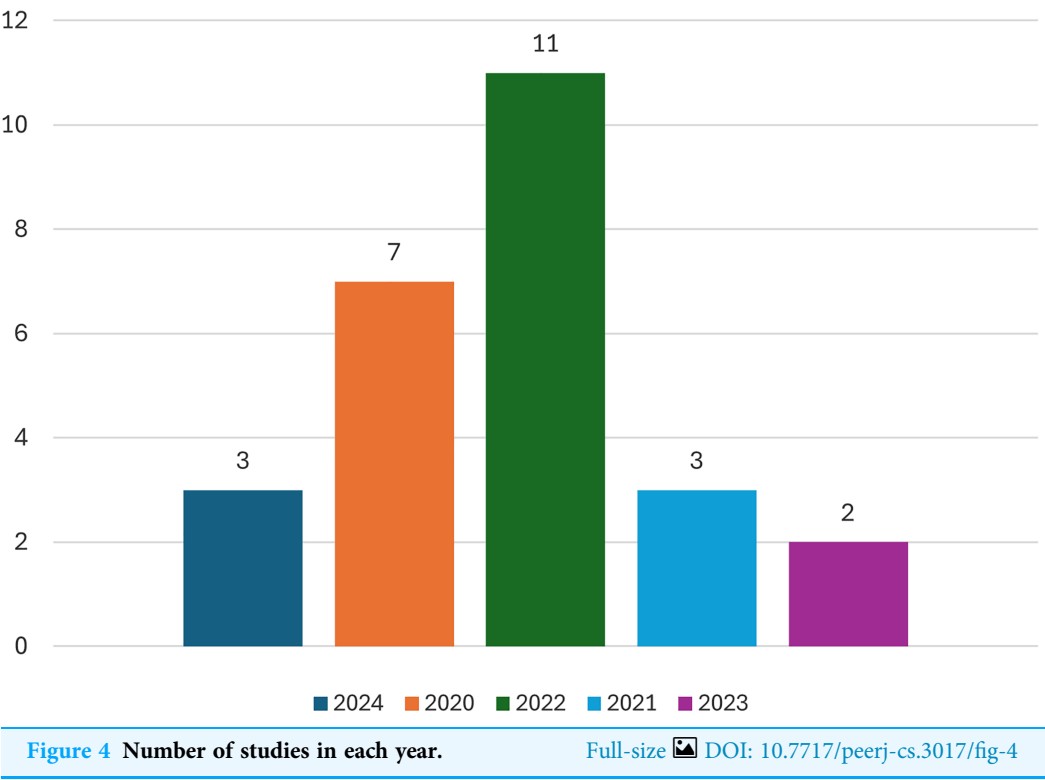

**Figure 4  Number of studies in each year.**

## FINDING OF SLR

Figure 3 illustrates the whole process for identifying articles in this research. A total of 1,028 studies were identified across various databases, with 738 originating from the Web of Science, 10 from IEEE, 118 from the ACM Digital Library, 41 from Taylor & Francis, and 122 from Google Scholar. subsequently, include these studies in Mendeley and eliminate the two duplicate studies. Following the initial screening, a total of 1,027 articles were identified based on their title and abstracts. Out of these, 821 studies were excluded since they did not match the inclusion criteria. A majority of the unmet criteria pertained to fields other than crime, amounting to a total of 596 studies.

Subsequently, the remaining 206 articles undergo a comprehensive evaluation of their full texts. Out of these, 179 articles were deemed ineligible based on the predetermined criteria and were therefore omitted from further analysis. The majority of the unmet criteria following the assessment of full-text eligibility lack the use of crime data. Ultimately, 27 articles met all the criteria needed and appeared appropriate to be included in this comprehensive systematic literature review for thorough analysis.

### Statistics of the included studies

This section will provide an overview of the statistical data from the studies that have been included. Figure 4 displays the number of publications for each year. Most of the research, specifically seven, was published in 2022, while the minority, only two, was published in 2023.
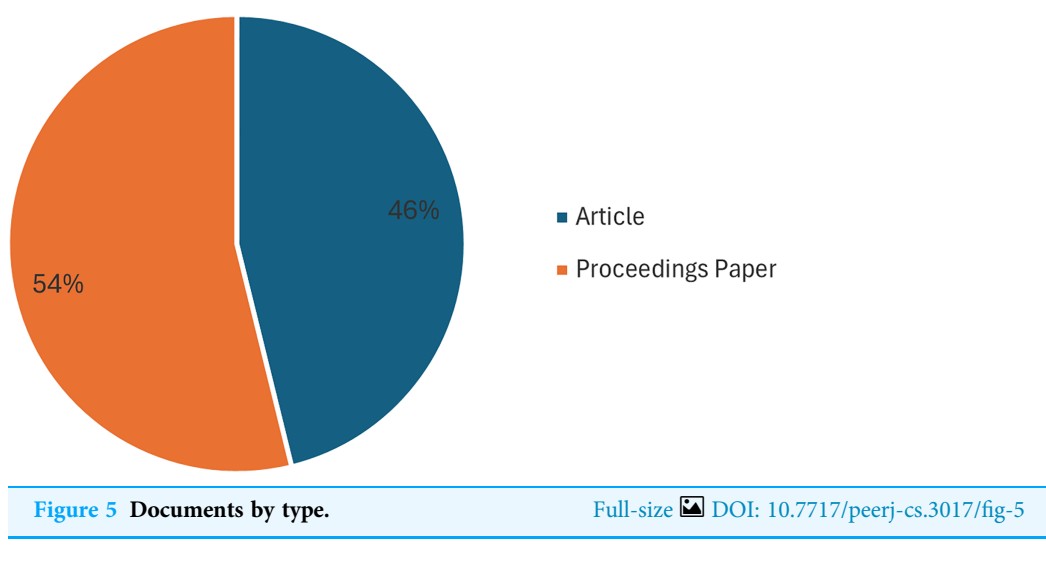

**Figure 5** Documents by type.               

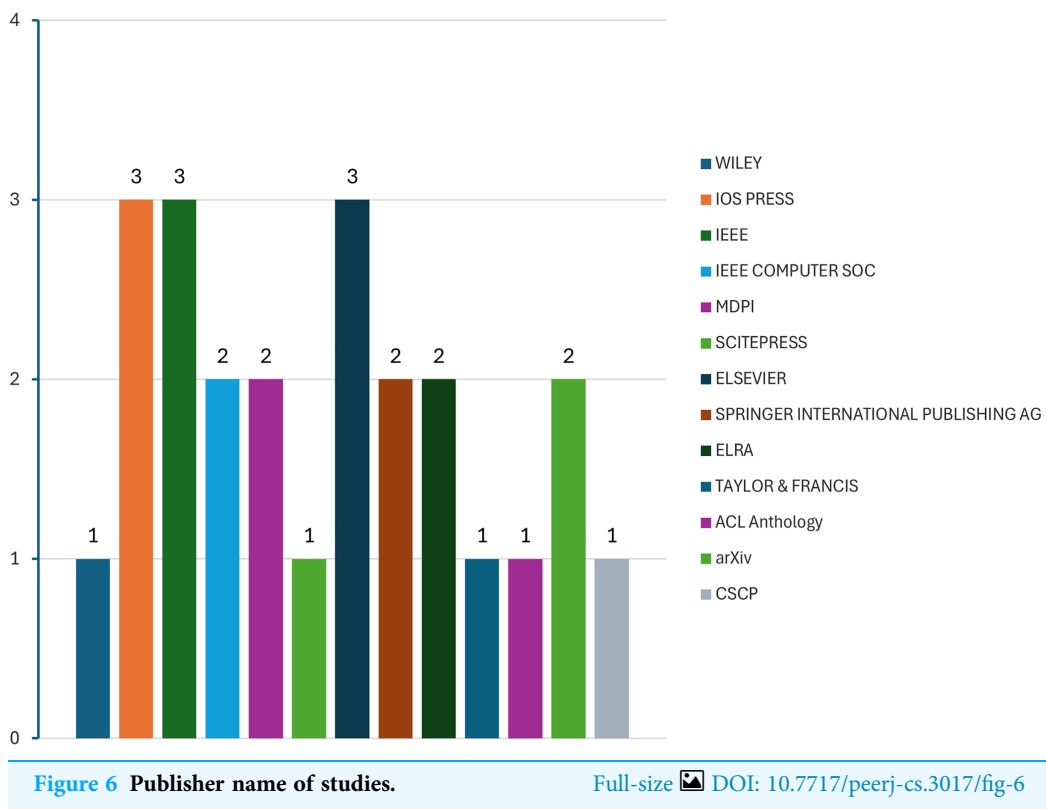

**Figure 6** Publisher name of studies.     

Figure 5 illustrates the document type for the investigations. The proceedings article comprises the majority of the studies included, with 14 (54%). The remaining studies are journal Articles, with 12 (46%).

Figure 6 shows the publisher's name for each study. IEEE, IOS Press, and Elsevier have the highest number of included studies, each with three.

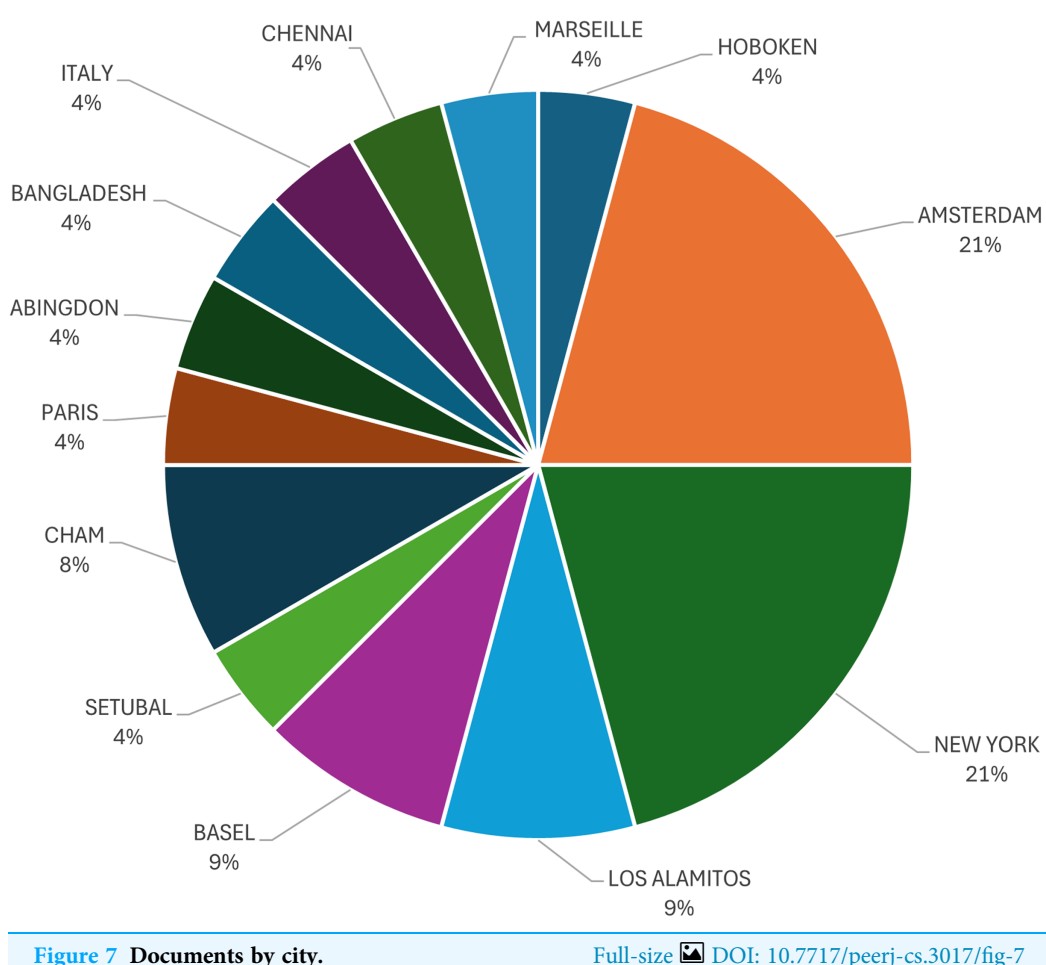

**Figure 7 Documents by city.**     

The final figure in the statistics is Fig. 7, which provides an overview of the studies conducted by the city. Amsterdam and New York have the most studies, with a total of five.

## The quality of the studies included

An assessment of the quality of the 27 studies was conducted. The general quality of the studies included was considered good since they all received a score of 9 or higher. Therefore, no studies were excluded after the quality assessments. Out of the total of 11 criteria, three studies (*Agarwal et al., 2022*; *Bugueno & Mendoza, 2020*; *Clavie & Alphonsus, 2021*) fulfilled every criterion, whereas 17 studies (*Akram, Shahzad & Bashir, 2023*; *Aluru et al., 2021*; *Cascavilla, Catolino & Sangiovanni, 2022*; *Cho & Choi, 2022*; *Ding & Yang, 2020*; *Djandji et al., 2020*; *El-Alami, Alaoui & Nahnahi, 2022*; *Jamil et al., 2023*; *Kapil & Ekbal, 2022*; *Kittask, Milintsevich & Sirts, 2020*; *Modha, Majumder & Mandl, 2022*; *Mubarak et al., 2020*; *Rahman et al., 2020*; *Shapiro, Khalafallah & Torki, 2022*; *Sharif & Hoque, 2022*; *Soldevilla & Flores, 2021*; *Yu, Li & Feng, 2024*; *Mozafari et al., 2024*; *Adam, Zandam & Inuwa-Dutse, 2024*) partially met some criteria, and 13 studies (*Akram, Shahzad & Bashir, 2023*; *Arcila-Calderón et al., 2022*; *Cascavilla, Catolino & Sangiovanni, 2022*; *Djandji et al., 2020*; *El-Alami, Alaoui & Nahnahi, 2022*; *Hossain et al., 2023*;

**Table 3 The quality assessment result.**

| The eleven criteria | Fully Met | Partially Met | Not Met |
|---|---|---|---|
| 1. Does the article rely on empirical research? | 27 | 0 | 0 |
| 2. Does the research have a well-defined set of aims? | 25 | 1 | 0 |
| 3. Does the research provide a sufficient explanation of the context in which it was conducted? | 27 | 0 | 0 |
| 4. Did the research design effectively address the objectives of the study? | 27 | 0 | 0 |
| 5. Did the recruitment strategy align with the objectives of the research? | 27 | 0 | 0 |
| 6. Was there a control group available for comparing treatments? | 6 | 14 | 7 |
| 7. Was the data acquired in a manner that effectively addressed the research issue? | 26 | 1 | 0 |
| 8. Was the data analysis conducted with enough rigor? | 12 | 4 | 11 |
| 9. Has the extent to which the link between the researcher and participants has been thoroughly evaluated? | 27 | 0 | 0 |
| 10. Does the report contain an unambiguous presentation of the research findings? | 27 | 0 | 0 |
| 11. Is the study of value more beneficial for research or practice? | 27 | 0 | 0 |

*Kittask, Milintsevich & Sirts, 2020*; *Mubarak et al., 2020*; *Rahman et al., 2020*; *Schirmer, Kruschwitz & Donabauer, 2022*; *Shapiro, Khalafallah & Torki, 2022*; *Soldevilla & Flores, 2021*; *Yu, Li & Feng, 2024*; *Cruz & Cheng, 2020*; *Ullah et al., 2024*) did not meet some criteria.

The criteria that were mostly partially met or unmet in studies were criteria 6 and 8 in terms of rigor. This is because most of the studies did not compare their findings with either previous studies or baseline models for criterion 6, and for criterion 8 data was unbalanced or had some inconsistencies. Table 3 contains a comprehensive explanation of all the criteria.

## Description of the model and classification

The investigations were divided into low and high resource categories. Low-resource languages are those that possess a limited number of resources and require further exploration, such as Urdu, Arabic, and Estonian. High-resource languages are those that possess substantial support and resources, such as English, Chinese, Japanese, and Spanish. This SLR comprises a total of 27 articles, with 17 categorized under low and 10 under high-resources languages.

This section provides comprehensive information on the model used in each study, including the year of publication (in chronological order), the model's name, the supported languages, the classification task, and the data type, as demonstrated in Tables 4 and 5 for the low-resource and high-resource languages, respectively.

The models employed in the low-resource category primarily include mBERT, with 14 studies supporting various languages. AraBERT is utilized in four studies, exclusively supporting language. Distlm-BERT is utilized in four studies, encompassing 104 languages. XLM-RoBERTa is utilized in five studies, encompassing 100 languages. Additionally, there is a single study for each of the following models. MarBERT which is specifically designed

**Table 4 Description of the model and classification for low-resource languages.**

| Articles | Year | Model name | Supported language | Classification task | Type of data |
|---|---|---|---|---|---|
| *Cruz & Cheng (2020)* | 2020 | mBERT, Distilm-BERT | 104 | Hateful content detection | Social media Tweets |
| *Rahman et al. (2020)* | 2020 | mBERT (uncased) | 104 | Domain classification | Documents |
| *Djandji et al. (2020)* | 2020 | AraBERT base | Arabic | Detecting offensive language | Social media Tweets |
| *Kittask, Milintsevich & Sirts (2020)* | 2020 | mBERT (cased), XLM-100 (cased), Distilm-BERT (cased), XLM-R base | 104 100 104 100 | Text classification | Paragraphs |
| *Mubarak et al. (2020)* | 2020 | AraBERT base mBERT | Arabic 104 | Detecting offensive language | Social media Tweets |
| *Aluru et al. (2021)* | 2021 | mBERT | 104 | Hate speech classification | Social media Tweets |
| *Arcila-Calderón et al. (2022)* | 2022 | mBERT | 104 | Racist and xenophobic hate speech classification | Social media texts |
| *Modha, Majumder & Mandl (2022)* | 2022 | mBERT, | 104 | Aggressive text detection | Social media Posts |
| *Sharif & Hoque (2022)* | 2022 | mBERT (uncased), Distilm-BERT (cased), Bangla-BERT base, XLM-R base | 104 104 Bengali 100 | Aggressive content detection | Social media texts |
| *Kapil & Ekbal (2022)* | 2022 | mBERT | 104 | Aggression, hate, and abuse detection | Social media texts |
| *Shapiro, Khalafallah & Torki (2022)* | 2022 | AraBERT base MarBERT | Arabic | Detecting offensive language | Social media Tweets |
| *El-Alami, Alaoui & Nahnahi (2022)* | 2022 | mBERT AraBERT base | 104 Arabic | Multilingual offensive language detection task | Social media Tweets |
| *Akram, Shahzad & Bashir (2023)* | 2023 | mBERT | 104 | Hateful content detection | Social media Tweets |
| *Hossain et al. (2023)* | 2023 | mBERT | 104 | Crime text classification and drug modeling | Bengali News Articles |
| *Ullah et al. (2024)* | 2024 | mBERT Distilm-BERT Multilingual Mini-LM XLM-R base | 104 100 | Cybercrimes text classification | Social media Tweets |
| *Mozafari et al. (2024)* | 2024 | mBERT XLM-R base ParsBERT ALBERT-Persian | 104 100 Persian | Detecting offensive language | Social media Tweets |
| *Adam, Zandam & Inuwa-Dutse (2024)* | 2025 | mBERT XLM-R | 104 100 | Detecting offensive language | Social media Posts |

for the Arabic language. Bangla-BERT, which was particularly designed for the Bengali language. The XLM-100 is designed to provide support for 100 languages. ParsBERT designed for the Persian language.

**Table 5 Description of the model and classification for high-resource languages.**

| Articles | Year | Model name | Supported language | Classification task | Type of data |
|---|---|---|---|---|---|
| *Ding & Yang (2020)* | 2020 | bert-base-chinese | Chinese | Criminal scenario | Burglary case descriptive files |
| *Bugueno & Mendoza (2020)* | 2020 | BERT | English | Online harassment detection | Social media Tweets |
| *Clavie & Alphonsus (2021)* | 2021 | LEGAL-BERT (*Chalkidis et al., 2020*) | English | Court decision | Documents |
| *Soldevilla & Flores (2021)* | 2021 | BERT-base-uncased | English | Gender violence classification | Social media messages |
| *Schirmer, Kruschwitz & Donabauer (2022)* | 2022 | BERT base (uncased) | English | Violent or non-violent text chunks classification | Documents from different tribunals |
| *Cascavilla, Catolino & Sangiovanni (2022)* | 2022 | BERT RoBERTa | English | Classifying illegal activities & drug types | Text-based dark web pages |
| *Cho & Choi (2022)* | 2022 | BERT, GAN-BERT | English | Topic classification | Documents from different newsgroups |
| *Agarwal et al. (2022)* | 2022 | BERT-base-uncased | English | Sexual predator detection task | Predator conversations files |
| *Jamil et al. (2023)* | 2023 | BERT RoBERTa | English | Detecting dangerous events | Wikipedia and text *corpus* |
| *Yu, Li & Feng (2024)* | 2024 | BERT RoBERTa | English | Topic classification | Documents from different newsgroups |

The models used in the high-resource category are primarily BERT-based (cased and uncased), with eight studies exclusively supporting the English language. Immediately following that. In three studies, RoBERTa is exclusively used to support the English language. Furthermore, there is a single study for each of the following models: Chinese_L-12_H-768_A-12, which is specifically designed for the Chinese language. LEGAL-BERT is used in a single study that is intended to address legal content in the English language. Finally, GAN-BERT is used in a single study that is intended for the English language and a generative adversarial setting.

The classification task for both categories varies depending on the content of the data, although all fall under the umbrella of illegal actions in broad terms. The data mostly consists of social media posts and text, with additional kinds including court or tribunal documents and newsgroup data.

## Description of the dataset

This section offers a thorough overview of the entire dataset process, encompassing details such as the language of the data, data source (whether it is available online or newly collected), number of categories (binary or multi), types of data categories (including various crime types), dataset size, labeling method (manual or pre-annotated), whether the dataset is balanced or not, the technique employed if the dataset is not balanced, the size of the dataset after balancing, and finally, the specifics of data splitting. All the details are outlined in Tables 6 and 7.

The majority of the dataset languages in the low-resources category belong to Arabic, accounting for 29.4% (five studies). Bengali follows with a rate of 17.6% (three studies), and then Hindi with a rate of 11.7% (two studies) and Urdu for Pakistan with a rate of 11.7%

**Table 6 Description of the dataset for low-resource languages.**

| Article | Language | Source | Number of categories | Categories of data | Size | Way of labeling | Balanced or no | Use a technique to balance the data or not | After balanced data | Splitting the data |
|---|---|---|---|---|---|---|---|---|---|---|
| Cruz & Cheng (2020) | Filipino | X Platform | 2 | -hate -non-hate | 10,000 | N/A | Relatively | N/A | – | N/A |
| Rahman et al. (2020) | Bengali | Open source 1-BARD, 2-OSBC 3-ProthomAlo | 1-5 2-11 3-6 | Include crime | 1-50,560 2-78,796 3-128,761 | Annotated | No | No | – | Train: 80% Test: 20% |
| Djandji et al. (2020) | Arabic | X Platform | 2 | Offensive Not offensive | 10,000 | Annotated | No | Yes, over/ undersampling | – | Train: 70% Test: 20% Development: 10% |
| Kittask, Milintsevich & Sirts (2020) | Estonian | Postimees Estonian newspaper | 4 | Include crime Topic: Negative Ambiguous Positive Neutral | 4,088 | Annotated with sentiment & with rubric labels. | No | No | – | Train: 70% Test: 20% Development: 10% |
| Mubarak et al. (2020) | Arabic | X Platform | 4 | - Offensive - Vulgar - Hate speech - Clean | 10,000 | Experienced annotator | No | No | – | N/A |
| Aluru et al. (2021) | 1. Indonesian 2. Polish 3. Arabic | Public Dataset From X platform | 2 | -Hate speech -Normal | 13,169 & 713 9,788 4,120 & 1,670 | Annotated | Yes No No & YES | No | – | Train: 70% Test: 20% Validation: 10% |
| Arcila-Calderón et al. (2022) | 1. Greek 2. Italian | PHARM datasets | 2 | -Racist/xenophobic hate tweets -Non-racist/ xenophobic hate tweets | 1-10,399 2-10,752 | Manual | Yes | – | – | N/A |
| Modha, Majumder & Mandl (2022) | Hindi | TRAC | 3 | Non-aggressive (NAG), Overtly Aggressive (OAG), & Covertly Aggressive (CAG) | 15,001 | Annotated | Yes | – | – | N/A |
| Sharif & Hoque (2022) | Bengali | BAD Facebook & YouTube | 1-2 2-4 | 1-AG & NoAG 2-ReAG & PoAG & VeAG & GeAG. | 14,443 | Manual | 1-Yes 2-No | 1- - 2-No | – | Train: 80% Test: 10% Validation: 10% |
| Kapil & Ekbal (2022) | Hindi | Eight datasets from social media | 1-2 2-3 | Hate, normal Hostile, non-hostile CAG, NAG, OAG Abusive, hate, natural | Different sizes | Annotated | Some are balanced and some not | No | – | Train: 80% Test: 20% |

(Continued)

| Article | Language | Source | Number of categories | Categories of data | Size | Way of labeling | Balanced or no | Use a technique to balance the data or not | After balanced data | Splitting the data |
|---|---|---|---|---|---|---|---|---|---|---|
| Shapiro, Khalafallah & Torki (2022) | Arabic | X Platform | 2 | Offensive / Not offensive | 12,700 | Annotated | No | No | – | Train: 70% Test: 20% Development: 10% |
| El-Alami, Alaoui & Nahnahi (2022) | Arabic | SemEval'2020 competition -Arabic dataset | 2 | 1 = Offensive 0 = Not offensive | 7,800 | Annotated | No | No | – | Train: 80% Test: 20% |
| Akram, Shahzad & Bashir (2023) | Urdu (Pakistan) | X Platform | 2 | Hatful / Neutral | 21,759 | Manual | No | No | – | Train: 90% Test: 10% |
| Hossain et al. (2023) | Bengali | Kaggle & Bangla Newspaper | 1-2 2-4 | 1-Crime & Others 2-Murder, Drug, Rape & Others | Approximately 5.3 million entries | Annotated & Manual | N/A | N/A | – | N/A |
| Ullah et al. (2024) | Urdu (Pakistan) | X Platform | 5 | Cyber Terrorism Hate Speech Cyber Harassment Normal Offensive | 7,372 | Three Manual Annotator | No | No | – | Train: 80% Test: 20% |
| Mozafari et al. (2024) | Persian | X Platform | 2 | Offensive / Not offensive | 6,000 | Manual | No | No | – | N/A |
| Adam, Zandam & Inuwa-Dutse (2024) | Hausa (Africa) | X & factbook Platforms | 2 | Offensive / Not offensive | N/A | Manual | N/A | N/A | – | N/A |
**Table 7 Description of the dataset for high-resource languages.**

| Article | Language | Source | Number of categories | Categories of data | Size | Way of labeling | Balanced or no | Use a technique to balance the data or no | After balanced | Splitting the data |
|---|---|---|---|---|---|---|---|---|---|---|
| Ding & Yang (2020) | China | The Criminal Investigation Corps of the Shanghai Public Security Bureau | 3 | N/A | 6,000 | Manual | No | Yes, Oversampling method | 9,000 | Train: 60% Test: 20% Validation: 20% |
| Bugueno & Mendoza (2020) | English | ECML/PKDD 2019 conference SIMAH | 1-2 2-4 | 1-harassment or non-harassment 2-Non harassment, sexual harassment, physical harassment & Indirect harassment | 10,622 | Annotated | No | Yes, SMOTE method | N/A | Train: 60% Test: 20% Validation: 20% |
| Clavie & Alphonsus (2021) | English | Public dataset 1-Overruling the European Convention of Human Rights (Chalkidis, Androutsopoulos & Aletras, 2019) 2-ECHR-Casetext, a company focused on legal research software | 1-2 2-2 & 66 | 1-N/A 2-Positive if any human rights article or protocol has been violated and negative otherwise. | 11,500 | Annotated | Yes | – | – | N/A |
| Soldevilla & Flores (2021) | English | Schrading's investigation (Schrading, 2015) 1-Reddit 2-X | 2 | violence & non-violence | 1-113,910 2-30,377 | Annotated | Yes | – | – | Train: 64% Test: 20% Validation: 16% |
| Schirmer, Kruschwitz & Donabauer (2022) | English | GTC The ECCC, the ICTR, & the ICTY | 2 | 1 = violence & 0 = no violence | 1,475 | Manual | Yes | – | – | Train: 80% Test: 10% Validation: 10% |

(Continued)

| Article | Language | Source | Number of categories | Categories of data | Size | Way of labeling | Balanced or no | Use a technique to balance the data or no | After balanced | Splitting the data |
|---|---|---|---|---|---|---|---|---|---|---|
| Cascavilla, Catolino & Sangiovanni (2022) | English | Duta10k Agora Dark & surface web | 19 | Include: drugs, substances for drugs, drug paraphernalia, violence, porno & fraud | 113,995 | Some manuals & some annotated | No | No | – | N/A |
| Cho & Choi (2022) | English | Kaggle 20 News Group | 20 | Include Guns | 18,846 | Annotated | No | Yes, in GAN-BERT No in BERT | N/A | Train: 60.04% Test: 39.96% |
| Agarwal et al. (2022) | English | PAN-2012 competition on the sexual predator identification task. | 2 | 1 = predatory & 0 = non-predatory | 32,092 | Annotated | No | Yes, give more weight to the minority class | – | Train: 30.4% Test: 69.6% |
| Jamil et al. (2023) | English | MAVEN Wikipedia | 3 | 1) Dangerous events; 2) Top-level dangerous events; 3) Sub-level dangerous events. | 21,412 | Annotated& Manual | No | Yes, the model is equally trained in each class | – | Train: 70% Test/Validation: 30% |
| Yu, Li & Feng (2024) | English | Kaggle 20 News Group | 20 | Include Guns | 18,846 | Annotated | No | No | – | Train: 60.03% Test: 39.97% Validation: 10% from train set |

(two studies), while the remaining languages, including Estonian, Italian, Greek, Polish, and Filipino, Persian and Hausa, each have a rate of 5.8% (one study) for each. On the other hand, the majority of dataset languages in the high-resources category are English, with 90% (nine studies), and Chinese, with 10% (one study).

Certain datasets for both categories are obtained from Kaggle, while others are not accessible online. The number of categories varies, with some being binary and others multi-categorical (ranging from three to 66 categories). Additionally, the size of the data varies. Among these, the smallest dataset consists of 1,475 entries, while the largest datasets contain millions of entries. However, the small datasets are labeled manually, whereas the largest dataset is pre-annotated.

Out of the total of 17 datasets in the low-resources category, 52.9% of them are unbalanced. Among these unbalanced datasets, 47% (eight datasets) do not employ any technique to balance the data, while just one study, 5.8% (one dataset), makes use of the over- and under sampling technique. Out of the total number of datasets, which is two, are balanced, and the remaining datasets are balanced and not balanced using a different dataset. Conversely, the high-resources category employs 10 datasets. Thirty percent (three datasets) are balanced, while 70% (seven datasets) are not. However, the unbalanced dataset is resolved using a variety of techniques, including GAN, weight, oversampling, and SMOTE.

When splitting the dataset in the low-resource category, 29.4% of the studies employed the method of train, test, and validation splitting. 29.4% of the studies employ train and test splitting, while the remaining studies do not disclose the splitting details. When employing the three splitting techniques, the splitting percentages are as follows: for training, 80%, and 70%; for testing, 20%, and 10%; and for validation, 10%. If the two-splitting technique is applied (train 90%, 80%, and test 20%, 10%). When employing the three splitting techniques in the high-resource category, the dividing percentages are as follows: 80%, 70%, and 60% for training, 20%, and 10% for testing, and 10% for validation. If the two-splitting technique is implemented (train 60%, test 30%, and train 40%, test 70%).

## Model evaluation

In this section, the evaluation of a model is initially determined by the type of evaluation metrics employed in each study. Secondly, in their publication, do they include a comparison with other ML, DL, or Transformer-based models? Thirdly, comparison with prior studies, and finally, the most robust result. These are detailed in Tables 8 and 9.

As demonstrated in the table below, the primary evaluation metric in both categories is the F1-score, which is utilized in 82.3% of the studies. The accuracy, precision, and recall metrics the second most used, appearing in 58.8% of studies. Other metrics, such as sensitivity, specificity, micro F1-scores, weighted F1-score (WF), macro F1-scores, error, training time, loss, computing power, confusion matrix, F0.5-score, and AUC.

When it comes to evaluating comparisons in studies, 82.3% (14 studies) in the low-resources category do not provide comparisons with previous studies, whereas 17.6% (three studies) do include comparisons. When it comes to comparing with their study,

**Table 8 Model evaluation for low-resource languages.**

| Article | Evaluation metrics | Compare the result with other models in their article or not | Compare the result with previous studies or no | Best result |
|---|---|---|---|---|
| Cruz & Cheng (2020) | Accuracy, Loss | No | No | mBERT outperforms Distilm-BERT with 74% |
| Rahman et al. (2020) | Precision, Recall, F1-score, Accuracy | Yes, ELECTRA | No | ELECTRA gets the best accuracy in all datasets |
| Djandji et al. (2020) | Macro-F1-score | No | No | AraBERT achieved 90% |
| Kittask, Milintsevich & Sirts (2020) | Accuracy | Yes, fastText | No | XLM-RoBERTa achieved the highest & DistilmBERT the lowest |
| Mubarak et al. (2020) | Precision, Recall, F1-score | Yes, fastText, SVM, Decision Tree, Random Forest, GaussianNB, Perceptron, AdaBoost, Gradient Boosting, Logistic Regression | No | AraBERT achieved the highest F1-score of 83%, while mBERT achieved 76%. |
| Aluru et al. (2021) | F1-score | Yes, MUSE + CNN-GRU Translation + BERT LASER + LR mBert | No | mBERT is superior in Arabic with a F1-score of 83% and in Indonesian with a F1-score of 81%. In Polish, the translation with bert is superior with a score of 71%. |
| Arcila-Calderón et al. (2022) | Accuracy and F1-score | No | No | mBERT achieves an accuracy rate of 91% in Italian and 81% in Greek. |
| Modha, Majumder & Mandl (2022) | Precision, Recall, F1-score, WF | Yes, 16 traditional & deep neural classifiers | No | CNN is better with a WF of 64% |
| Sharif & Hoque (2022) | Precision, Recall, F1-score, Error, WF | Yes, LR, RF, NB, SVM, CNN, BiLSTM & CNN + BiLSTM | Yes | In WF metrics The two-class XLM-RoBERTa is the Best In four classes, Bangla-BERT is the best Outperforms previous studies |
| Kapil & Ekbal (2022) | Weighted-F1 | Yes, MuRIL, M-BERT-Bilstm, MuRIL-Bilstm, and cross-lingual information. | Yes | The best model is cross-lingual information with a rate of 95% |
| Shapiro, Khalafallah & Torki (2022) | Precision, Recall, F1-score, Accuracy | Yes, baseline models | No | MarBERTv2 outperforms AraBERT and other baseline models with an 84% F1-score and 86% accuracy |
| El-Alami, Alaoui & Nahnahi (2022) | Accuracy and F1-score | Yes, CNN, RNN, bidirectional RNN, ULMFiT, ELMo, SVM & combined models | No | AraBERT achieves the highest F1-score of 93% in Arabic, surpassing the models it was compared against, and 91% accuracy. |
| Akram, Shahzad & Bashir (2023) | Precision, Recall, F1-score, Accuracy | Yes, NB, SVM, LR, RF, CNN, LSTM and BiLSTM | No | mBERT is the most effective model, obtaining an F1-score of 0.83%. |
| Hossain et al. (2023) | Precision, Recall, F1-score | No | No | mBERT in Crime classification achieved 96% Precision, and crime type achieved 98% recall. |

| Article | Evaluation metrics | Compare the result with other models in their article or not | Compare the result with previous studies or no | Best result |
|---|---|---|---|---|
| Ullah et al. (2024) | Precision, Recall, F1-score, Accuracy | No | No | XLM-R had the best accuracy with 77% |
| Mozafari et al. (2024) | Precision, Recall, F1-score, Accuracy, F1 Macro | Yes, SVM, CNN, BiLSTM + CNN | Yes | SVM gets the best F1 Macro then ParsBERT |
| Adam, Zandam & Inuwa-Dutse (2024) | Precision, Recall, F1-score, Accuracy | Yes, SVM, NB, XGBoost, CNN | No | mBERT has the best performance with 86% accuracy and 89% Precision among models |

**Table 9 Model evaluation for high-resource languages.**

| Article | Evaluation metrics | Compare the result with other models in their article or no | Compare the result with previous studies or no | Best result |
|---|---|---|---|---|
| Ding & Yang (2020) | Precision, Recall, F1-score | Yes, Fast Txt, Text CNN, TextRNN, &TextRNN with Attention | No | The BERT model has the best result of all the metrics. Precision: 98.52%, Recall: 98.50%, F1-score: 98.50% |
| Bugueno & Mendoza (2020) | F1-score, Accuracy | Yes, CNN1 CNN2 RNN1 RNN2 RNN3, RF, Linear SVM, & Gaussian SVM | Yes | Before balancing the data, models are less accurate than Gaussian SVM, after balanced data BERT does not achieve the best outcome. |
| Clavie & Alphonsus (2021) | Accuracy | Yes, SVM | Yes | Legal-BERT from the previous Studies is the best performance |
| Soldevilla & Flores (2021) | AUC, Sensitivity, Specificity, Accuracy | No | No | BERT gets the best AUC with 0.9603 on 5e-5 LR & two epochs |
| Schirmer, Kruschwitz & Donabauer (2022) | Precision, Recall, Micro F1-scores, Macro F1-scores | No | No | BERT in A mixed dataset effectively predicts if a text passage from one of three genocide trials contains witness reports of violence with a 0.81 macro F1-score |
| Cascavilla, Catolino & Sangiovanni (2022) | Accuracy, Confusion matrix, Precision, Recall, Training time, Computing power | Yes, RoBERTa, LSTM & ULMFit | No | BERT surpasses all models in major class and drug type accuracy with 96.08% and 91.98%, respectively. |
| Cho & Choi (2022) | Accuracy | Yes, LMGAN | No | GAN-BERT outperforms BERT and LMGAN. |
| Agarwal et al. (2022) | Precision, Recall, F1, and F0.5-score | Yes, RoBERTa and BertForSequenceClassification, RoBERTa frozen & RoBERTa tuned | Yes | BERT outperforms all models except relative to previous research in precision metrics. |

(Continued)

| Article | Evaluation metrics | Compare the result with other models in their article or no | Compare the result with previous studies or no | Best result |
|---|---|---|---|---|
| Jamil et al. (2023) | Precision, Recall, F1-score, Accuracy | Yes, XLNet | No | Regarding accuracy, BERT performs better in the most dangerous events, while RoBERTa and XLNet perform better for limited tasks. |
| Yu, Li & Feng (2024) | Accuracy | Yes, BERT & RoBERTa | No | AM-RoBERTa outperforms all the models with an accuracy of 90.32% |

70.5% of studies compare, whereas 23.07% do not. However, 30% (three studies) in the high-resources category do not provide comparisons with previous studies, whereas 70% (seven studies) do include comparisons. In terms of comparison with their study, 80% of studies do so, while 20% do not.

## Model hyperparameters

This section presents the hyperparameters derived from the author's specifications in each study, comprising the hidden size, learning rate, batch size, epoch number, and model name. Details are outlined in Tables 10 and 11.

Regarding Table 7, the epoch number is a value that falls inside a range, which can be a tiny number (3–8), or a medium number (16–20). The batch size options are 12, 16, and 32. The learning rates most utilized are 2e-5 and range between 2e-5 and 5e-6.

In Table 8, the epoch number for tiny (3–8) medium (10–20), or large (20–60). The batch size options are 1, 16, 32, 128, and 256. The learning rates most utilized are 2e-5 and range between 1e-4 and 5e-6. The study's authors do not provide information about the available padding sizes, but some of them use sizes of 256 and 512.

The batch size does not surpass 32, and the number of epochs is moderate for low-resource languages, nevertheless, for high-resource languages, the batch size can reach 256, and the epochs can extend to 60. On the other hand, the learning rates are comparable, except that the high-resource languages exhibit more diversity.

## DISCUSSION

The application of Transformer BERT in crime classification is constrained, as seen in Fig. 3. Most studies were excluded during the screening process based on their title and abstract. However, while seeing Fig. 4, it is evident that most studies experienced a rise in 2022. This systematic investigation focuses on the period until May 2025, indicating an emerging interest in the research domain. The limited number of studies included in the analysis, specifically 27 studies, demonstrates a need for further research in the field of crime. However, these studies contribute to addressing the study questions stated in subsection "Defying Research Question".

**Table 10 Model hyperparameters for low-resource language.**

| Article | Model name | Epoch number | Batch size | Learning rate | Hidden size |
|---|---|---|---|---|---|
| *Cruz & Cheng (2020)* | mBERT Distilm-BERT | 3 | 16 32 | 1e-5 | 768 |
| *Rahman et al. (2020)* | mBERT (uncased) | 20 | 16 | N/A | 768 |
| *Djandji et al. (2020)* | AraBERT base | 5 | 32 | N/A | 768 |
| *Kittask, Milintsevich & Sirts (2020)* | mBERT (cased), XLM-100 (cased), Distilm-BERT (cased), XLM-R base | (8–16) | N/A | (5e-5, 3e-5, 1e-5, 5e-6, 3e-6) | 768 1,024 768 768 |
| *Mubarak et al. (2020)* | mBERT AraBERT base | 30 | N/A | 5e-1 | 768 |
| *Aluru et al. (2021)* | mBERT | (1–5) | 16 | (2e-5, 3e-5, 5e-5) | 768 |
| *Arcila-Calderón et al. (2022)* | mBERT, | 3 | N/A | 3e-5 | 768 |
| *Modha, Majumder & Mandl (2022)* | mBERT | 3 | 32 | 2e-5 | 768 |
| *Sharif & Hoque (2022)* | mBERT (uncased), Distilm-BERT (cased), Bangla-BERT base, XLM-R base | 20 | 12 | 2e-5 | 768 |
| *Kapil & Ekbal (2022)* | mBERT | 2 | 30 | 2e-5 | 768 |
| *Shapiro, Khalafallah & Torki (2022)* | AraBERT base MarBERT | 100 with an early stopping patience of 10 | N/A | 2e-5 | 768 |
| *El-Alami, Alaoui & Nahnahi (2022)* | AraBERT base | 5 | 32 | N/A | 768 |
| *Akram, Shahzad & Bashir (2023)* | mBERT | 3 | 32 | N/A | 768 |
| *Hossain et al. (2023)* | mBERT | N/A | N/A | N/A | 768 |
| *Ullah et al. (2024)* | mBERT Distilm-BERT Multilingual Mini-LM XLM-R base | N/A | N/A | N/A | 768 384 |
| *Mozafari et al. (2024)* | mBERT XLM-R base ParsBERT ALBERT-Persian | 3 | 16 | 2e-5 | 768 |
| *Adam, Zandam & Inuwa-Dutse (2024)* | mBERT XLM-R base | N/A | N/A | N/A | 768 |

## Low-resources languages

This part will go over and discuss the four questions presented in subsection "Defying Research Question" regarding low-resource languages.

**Table 11 Model hyperparameters for high-resource language.**

| Article | Model name | Epoch number | Batch size | Learning rate | Hidden size |
|---|---|---|---|---|---|
| Ding & Yang (2020) | bert-base-Chinese | 20 | 128 | 5e-5 | 768 |
| Bugueno & Mendoza (2020) | BERT | 60 | 32 | N/A | 768 |
| Clavie & Alphonsus (2021) | LEGAL-BERT (Chalkidis et al., 2020) | 40 | N/A | 1e-4 | 768 |
| Soldevilla & Flores (2021) | BERT-base-uncased | From (2, 4) | 32 | 5e-5, 3e-5 & 2e-5 | 768 |
| Schirmer, Kruschwitz & Donabauer (2022) | BERT | 3 | 16 | N/A | 768 |
| Cascavilla, Catolino & Sangiovanni (2022) | BERT RoBERTa | 5 & 3 | N/A | 2e-4 & 2e-5 | 768 |
| Cho & Choi (2022) | BERT, GAN-BERT | 15 | N/A | 5e-6 | 768 |
| Agarwal et al. (2022) | BERT-base-uncased | N/A | 1 | 1e-5 | 768 |
| Jamil et al. (2023) | BERT RoBERTa | 10 | 16 | 2e-5 | 768 |
| Yu, Li & Feng (2024) | BERT RoBERTa | 3–4 10–20 | 32 128 or 256 | Among 5e-5, 4e-5, 3e-5, and 2e-5 from 1e-5 to 5e-5 | 768 |

**What are the most common BERT-based models used in the field of crime in Arabic and other low-resource languages?**

Based on Table 4, a total of eight models were utilized in 17 studies for the classification of texts within the crime domain. These models include the original mBERT version as well as various specialized models that have been designed based on BERT to meet certain purposes or languages. The models will be ranked from most to least in usage.

**– mBERT (cased & uncased)**

The "m" in mBERT stands for multilingual; it was first introduced in the article (Devlin et al., 2018), indicating that it has been pre-trained in a wide range of languages, up to 104 languages, to enable linguistic diversity. The two models, the cased and the uncased come from the same BERT basis. The parameters are 110M. The model was used in 11 studies: two for uncased, one for cased, and the remaining unspecified.

**– AraBERT**

Ara, the abbreviation for Arabic, is a model specifically developed for the Arabic language, introduced in the article (Antoun, Baly & Hajj, 2020). The model has two variations as the original BERT: base and large. Additionally, it includes two developmental versions: the original AraBERT v0.1/v1 and the subsequent AraBERT v0.2/v2. The parameters for the base model are 136M, whereas the large model has 371M. AraBERT base was used in four studies.

**– Distlm-BERT**

Distl refers to the distilled version of multilingual BERT, which supports 104 languages and maintains the same basic architecture as BERT, first presented in the article (*Sanh et al., 2019*). However, it preserves the characteristics of the original BERT, being smaller, cheaper, lighter, and faster, achieved by minimizing the model's size while preserving its capabilities. It contains 40% fewer parameters than BERT, operates 60% more rapidly, and maintains 97% of BERT's performance. The total parameters of Distlm-BERT are 134M. The Distlm-BERT cased model was used in three of the studies, but the uncased variant was not used.

**– XLM-R**

Cross-lingual Language Model-RoBERTa is an enhanced version of RoBERTa that supports 100 languages. First presented in the article (*Conneau et al., 2019*), it has two variations: XLAM-R base and XLAM-R large as the original BERT, pre-trained on 2.5 TB of filtered CommonCrawl data. XLAM-R base contains 250M parameters, while XLAM-R large contains 560M (*Goyal et al., 2021*). The XLAM-R base model was used in two studies.

**– Bangla-BERT**

This model has been designed to handle only the Bangalia language and was originally introduced in article (*Bhattacharjee et al., 2021*) . The parameters total 110M. The model was used only in one study.

**– XLM-100**

XLM-100 stands for Cross-lingual Language Model Pretraining that supports 100 languages first introduced in the article (*Conneau & Lample, 2019*). This model has 570M parameters and was used once among studies.

**– MarBERT**

MarBERT is a large-scale pre-trained masked language model that is dedicated to both Dialectal Arabic (DA) and Modern Standard Arabic MSA. It is one of three models introduced in the article (*Abdul-Mageed, Elmadany & Nagoudi, 2020*), and is supported by only Arabic. It has a total of 163M parameters.

**– ParsBERT**

ParsBERT is a monolingual BERT-based model specifically pre-trained for the Persian (Farsi) language. It is designed to handle Persian's unique linguistic features. It uses the BERT-base architecture (similar to the original BERT) but is trained on a large *corpus* of Persian texts.

To recap, the most frequently employed model is mBERT, as it is capable of supporting multiple languages. However, the low-resource language should not be entirely reliant on this model. It is necessary to investigate the models that are specifically designed for specific languages and to compare the more effective models.

Arabert is the Arabic model that is most frequently employed, followed by Marbert. The other models that support Arabic require further investigation. Arabert demonstrated superiority over mBERT in studies (*Mubarak et al., 2020*; *El-Alami, Alaoui & Nahnahi,*

2022), and MarBERT outperformed Arabert in an additional study (*Shapiro, Khalafallah & Torki, 2022*). Therefore, it is necessary to conduct a study to compare the two models.

Urdu and Bangla are the second most frequently used languages after Arabic. Even though Bangla, the mBERT model was employed in two of the studies, while one study employed the Bangla version. Therefore, to conduct a comparison, Bangli must conduct additional research on their model, as the mBERT model performed the worst in the study (*Rahman et al., 2020*).

The mBERT models are also utilized in other languages, including Urdu, Estonian, Persian, Filipino, Hausa, and Polish. Therefore, it is necessary to investigate the models that are specifically designed for their respective languages.

### In crime-related text data, does the use of BERT-based models enhance accuracy compared to ML, DL, and other Transformer models?

According to the data shown in Table 8, 14 out of 17 studies concluded that BERT-based models outperformed other ML and DL models and improved accuracy across various datasets, except for three studies. The studies will be discussed and ranked from the best to the worst results.

The initial study in the Arabic language, conducted by *Mubarak et al. (2020)*, aimed to identify offensive language through social media tweets. The article's primary contribution is the construction of a dataset comprising 10,000 tweets annotated with experiential annotators. The AraBERT and mBERT models are utilized to identify offensive tweets, and their performance is compared to several machine learning and deep learning models. The results indicate that AraBERT outperforms all the ML, DL models, and mBERT, achieving an F1-score of 83%. The obstacle is that the sample is unbalanced, and the F1-score improves by employing balancing techniques.

Using the same dataset from the authors (*Mubarak et al., 2020*), the second study for detecting offensive social media tweets employs the AraBERT by *Djandji et al. (2020)*. They employ balance techniques that involve over- and under-sampling. The results indicated that AraBERT has a 90% F1-score, which is superior to the result of *Mubarak et al. (2020)*. However, they are not comparable to any of the preceding results or ML and DL models. Their findings suggest that balanced data is beneficial for the precise detection of offensive tweets.

A further study by *Shapiro, Khalafallah & Torki (2022)*, employs the same models as prior research, AraBERT, and incorporates other Arabic models, namely MarBERT, to identify offensive tweets. The dataset comprises 12,700 imbalanced tweets. In comparison to baseline models, the results indicate that MarBERT outperforms AraBERT and baseline models, achieving an F1-score of 84%. The outcome surpasses that of *Mubarak et al. (2020)*, despite the dataset's imbalance, showing that MarBERT outperforms AraBERT by 1%. However, when addressing the imbalance, the findings of *Djandji et al. (2020)*, provide a superior performance with a 6% improvement. It is necessary to balance tweets to assess the efficacy of MarBERT in comparison to AraBERT.

The last study in the Arabic language (*El-Alami, Alaoui & Nahnahi, 2022*), was conducted using AraBERT and mBERT to Arabic offensive tweets. The class is binary, and

the dataset is pre-annotated with a range of 5,000 to 7,000 entries. The issue at hand is the presence of unbalanced data, which has not been effectively addressed. They compare the ML and DL models but do not compare them to previous studies. The study indicates that AraBERT outperforms the other models, including mBERT with an F1-score of 93% which is better than all previous Arabic studies. From one perspective, the researchers utilized an English dataset and then translated it into Arabic. This implies that there is currently no existing dataset available in Arabic and therefore, there is a need to generate one. Additionally, evaluate the balance of the data to determine if it affects the findings.

On top of that, it is important to note that there is a lack of comparisons with previous studies, which may be due to the limited availability of studies in this specific field. This factor should be taken into consideration. An additional consideration is that, when balancing the dataset, one must assess the accuracy differences between MarBERT, AraBERT, and other Arabic models.

The language with the most studies following Arabic is Bangla. According to a study (*Rahman et al., 2020*), learning Bengali does not increase the accuracy of using BERT basis models in illegal activity classification. However, the two remaining studies did show improvement (*Sharif & Hoque, 2022*; *Hossain et al., 2023*).

The initial study used four BERT-based models. Out of the four, three models can handle many languages, and one model, known as Bangla-BERT, is specifically developed to work with the Bengali language. To detect aggressive content in social media text. The data set was classified into binary and multi-class, with some of them being balanced and others not. However, the issue of imbalance was not resolved. They employed a ternary distribution. Regarding the outcome, they evaluate ML and DL models in comparison with earlier research. XLM-RoBERTa and Bangla-BERT outperform other models in WF metrics for binary and multi-class, respectively. It has been found that Bengali-BERT performs the best when there are more classes present. Study (*Rahman et al., 2020*), can utilize the Bangla-BERT version to improve their accuracy compared to ELECTRA rather than mBERT.

The mBERT model is employed in two studies by *Akram, Shahzad & Bashir (2023)* and *Ullah et al. (2024)* for the detection of hateful and cybercrime content on social media in the Urdu language. One study employs an additional model, XLM-R, without balancing the data. The results of the first study indicate that the mBERT model is superior to various ML and DL models, while the second study demonstrates that XLM-R is superior to mBERT.

In the other study (*Hossain et al., 2023*), Zero-Shot Classification was employed using the BERT base case to classify crime text and drug modeling in Bengali news articles. The dataset is extensive, consisting of over 5 million entries, some of which are pre-annotated and others that were manually annotated. A weakness of this study is the absence of any statement regarding data balance or the distribution of data splitting. Additionally, do not compare their findings with earlier research or with models from their study. However, their findings demonstrated that BERT significantly improves accuracy in crime classification, achieving a precision of 96% and a recall of 98% for different crime types.

Furthermore, the study (*Kittask, Milintsevich & Sirts, 2020*) employed BERT-based models, namely mBERT, XLM-100, XLM-R, and Distilm-BERT, to conduct experiments in the Estonian language. These models were utilized to classify newspaper text into four categories. Although the dataset size is adequate, a limitation of the study is that the dataset is unbalanced. The findings demonstrate that XLM-R outperforms ML and other transformation models, confirming it as the superior model.

Four additional studies will be discussed, all employing mBERT to detect aggressive, hateful, racist, and xenophobic content across several languages. The study (*Aluru et al., 2021*), evaluates various languages, including Indonesian, Polish, and Arabic. The dataset is balanced for Indonesian languages but unbalanced for Polish. Arabic employs two datasets: one is balanced, while the other is unbalanced. They conduct a comparison of various ML and DL models. The outcome for Arabic is 83%, surpassing the findings of the study (*Mubarak et al., 2020*) by an increase of 7%. However, the study (*El-Alami, Alaoui & Nahnahi, 2022*) demonstrated an 8% increase, resulting in a 91% performance in the mBERT model. The Indonesian achieved 81%, while the Polish recorded the lowest performance among the languages at 71%. The Polish language needs further exploration and enhancement. Nevertheless, all are superior to ML and DL models.

The study (*Arcila-Calderón et al., 2022*), employs the same previous model but for Greek and Italian, utilizing a balanced dataset comprising around 11,000 manually annotated tweets for both languages. The results indicate that the accuracy for Italian is 91% and for Greek is 81%, with no comparison to any previous findings or baseline models. Further exploration of these languages is necessary to find the differences or to employ BERT-based models specifically created for these languages.

The study (*Kapil & Ekbal, 2022*) utilizes eight datasets of the Hindi language, varying in size, with some being balanced and others not. The results are compared with previous and other models as stated in Table 8. Their model achieved superior accuracy, reaching 95%. Additionally, *Akram, Shahzad & Bashir (2023)* investigate the model on Urdu languages of Pakistan using around 22,000 manually annotated tweets, which are unbalanced and unresolved. They compare their results with baseline models of ML and DL, without referencing previous work. The results indicate that mBERT is the most effective language model, with 81% accuracy. One constraint is the imbalance that must be addressed to achieve further improvement in the detection of Hateful content.

The study (*Adam, Zandam & Inuwa-Dutse, 2024*), employs mBERT and XLM-R to classify offensive language in Hausa tweets; however, the dataset is not available for analysis. Their results demonstrate the superior performance of mBERT compared to several ML and DL methods.

The study (*Cruz & Cheng, 2020*) used mBERT and DistilBERT to classify hateful content in tweets written in Filipino, utilizing a dataset of 10,000 relatively balanced tweets. The findings indicate that mBERT outperforms DistilBERT, achieving an accuracy of 74%. The disparity is not being compared to any ML or DL model.

The remaining three studies *Rahman et al. (2020)*, *Mozafari et al. (2024)*, and *Modha, Majumder & Mandl (2022)*, one in the Bengali, one in Persian, and one in the Hindi languages, which contradicts whether BERT-based models enhance classification accuracy.

In the study (*Rahman et al., 2020*), mBERT was employed to categorize documents written in the Bangla language. The study also compared mBERT and another Transformer model known as ELECTRA. The study utilized three datasets, each with varying categories: 5, 11, and 6. The datasets were large. However, a notable limitation in this study was the unbalanced of the dataset, which was not addressed using different techniques. The study's findings demonstrate that ELECTRA outperforms mBERT in terms of accuracy across all datasets.

The study (*Mozafari et al., 2024*), utilizes mBERT, XLM-R base, ParsBERT, and ALBERT-Persian to classify offensive language in social media, analyzing 6,000 unbalanced tweets. A comparison using several ML and DL approaches is presented in Table 8. The results show that SVM performs better than ParsBERT.

The last study that does not obtain the highest accuracy utilizing the BERT-based model is the Study (*Modha, Majumder & Mandl, 2022*). In this study, they utilize the mBERT model to categorize aggressive tweets in Hindi into three categories. The dataset is large, but it lacks information regarding its balance. This absence of specifics restricts the ability to determine whether the issue in this study is due to unbalanced data or not. Additionally, the data-splitting process is missing. The mBERT is compared with 16 ML and DL models. The outcome demonstrated that the DL CNN model achieved the highest level of accuracy, reaching 64%, although it is not quite accurate.

### What are the most common evaluation metrics used in comparison?

As shown in Table 8, the F1-score accuracy, precision, and recall are the most frequently used metrics. Additionally, precision and recall are also considered, making it appropriate for comparison with previous studies. To achieve an accurate comparison, it is necessary to employ the same evaluations that are usually used in studies.

### Is there enough data available in the crime field?

According to the distribution of studies by language, 24% were conducted in Arabic, 12% in Bangla, Urdu, and Hindi, and 6% each in Estonian, Filipino, Persian, Hausa, Indonesian, Polish, Greek, and Italian. These languages necessitate additional labeled crime data due to their limited resources and the necessity for further dataset development. It is advisable to collect new data from various social media platforms, including X, Facebook, Instagram, and YouTube, and to use an annotation procedure. However, this process can be time-consuming. An alternative solution is to use existing data and enhance it by using data augmentation or any other technique, such as translating from different languages and combining the dataset.

## High-resources languages category

This part will address and discuss the four questions presented in subsection "Defying Research Question" regarding high-resource languages.

### What is the most common BERT model used in the field of crime in Arabic and other languages?

Based on Table 5, a total of five models were utilized in 10 studies for the classification of texts within the crime domain. These models include the original BERT as well as various specialized models that have been designed based on BERT to meet certain purposes or languages. The models will be ranked from most to least in usage.

**– BERT base (cased & uncased)**

The BERT basic model was first presented in the article (*Devlin et al., 2018*). It had pre-training exclusively in the English language, employing the MLM objectives. The BERT base model comes in two versions: cased and uncased. Cased refers to being case sensitive, which implies that "English" is distinct from "english." Uncased is the opposite of cased. The BERT-base (uncased & cased) has a total of 110M parameters, while the BERT-large (uncased & cased) version has 340M (*Devlin et al., 2018*).

This model was the most frequently utilized in studies, with a total of eight studies. Among these, three studies utilized a BERT-base uncased, and in five studies, it was not mentioned whether the model was cased or uncased.

**– RoBERTa**

RoBERTa is an abbreviation for A Robustly Optimized BERT approach, which was introduced and developed by researchers from Facebook AI in the article (*Liu et al., 2019*). It is a pre-trained model specifically designed for English and is available in two versions: cased and uncased. RoBERTa differs from the BERT base by incorporating alterations in important hyperparameters. These modifications involve the removal of pretraining objectives related to predicting the following sentence, as well as training with larger mini-batches and learning rates. The objective of optimization is to reduce the time taken for pre-training. The parameters count for the RoBERTa base is 125M, and for RoBERTa large it is 355M. The model used in three studies, the cased and uncased, was not specified.

**– bert-base-chinese**

This model has been designed to handle only the Chinese language and was originally introduced in article (*Devlin et al., 2018*). The parameters total 110M. The model was used only once.

**– LEGAL-BERT**

Legal BERT was designed specifically to target the legal domain, facilitating legal natural language processing, computational law, and legal technology applications developed by the authors of the article (*Chalkidis et al., 2020*). It has been pre-trained exclusively in the English language using data collected from various sources. It is regarded as a lightweight model, comprising 33% of the BERT base size, with a portion pre-trained from scratch on legal data, and it operates approximately four times faster. This model was used in a single study.

#### – GAN-BERT

GAN refers to generative adversarial networks; thus, GAN-BERT is a combination of GAN and BERT for semi-supervised learning and NLP tasks. It serves as a robust tool when data resources are limited and labeled data is few. This technique improves text and topic classification. GAN-BERT was used in one study (*Croce, Castellucci & Basili, 2020*).

### In crime-related text data, does the use of BERT-based models enhance accuracy compared to ML, DL, and other Transformer models?

According to the data shown in Table 9, 9 out of 10 studies concluded that BERT-based models outperformed other ML and DL models and improved accuracy across various datasets, except for one study.

The first study in Chinese by *Ding & Yang (2020)*, they utilized the original Chinese BERT and other DL models mentioned in the tables above to classify criminal Chinese files. They utilize three classification categories on an adequate-size dataset and address the imbalanced data problem by implementing the oversampling technique. Chinese BERT models demonstrated superior performance over DL models across all the metrics.

The first study in English (*Clavie & Alphonsus, 2021*), utilized LEGAL-BERT, a model that is designed for legal English text, to categorize court judgment documents. They utilized two datasets consisting of binary and multi-categories. The dataset is both large and equally distributed. Nevertheless, the drawback lies in the lack of description regarding the split of the training model. LEGAL-BERT outperforms BERT in the classification of legal content. The LEGAL-BERT needs to be trained to cover more languages.

The second study (*Soldevilla & Flores, 2021*), classifies social media gender violence. BERT-based uncased models and binary classification are used on X and Reddit data. The over 30,000-entry dataset was previously annotated and balanced. It follows a three-splitting distribution. The BERT model scored 96% AUC. The lack of comparisons with their own or previous studies is a restriction of this study.

The third study (*Schirmer, Kruschwitz & Donabauer, 2022*), addresses the classification of documents from various tribunals utilizing BERT base in binary classification. The study utilizes a small dataset consisting of 1,475 documents that have been manually classified and it is balanced. Three-splitting distribution is employed. Their macro F1-score reached 81%. However, the same study (*Soldevilla & Flores, 2021*), provided no comparisons to the previous study or their own.

The fourth study (*Cascavilla, Catolino & Sangiovanni, 2022*) classifies the illegal activities and drug types of dark web text. The researchers employed multi-classification on a dataset of over 100,000 instances of 19 distinct categories. Some of the instances were pre-annotated, while the remaining ones were manually annotated. The distribution of the splitting data is not present, and the dataset lacks balance, which fails to address the issue of unbalanced data. The outcome has shown that the BERT base model outperformed the DL model (LSTM) and transfer learning (ULMFit, RoBERTa) when compared with an accuracy of 96% and 92% for illegal activities and drug types respectively. Nevertheless, the absence of comparative previous studies hinders a thorough examination of the study's impact on the field.

The fifth study (*Cho & Choi, 2022*), was done by employing BERT-based uncased and GAN-BERT models, aimed to classify the topic of newspaper documents. The researchers employed a multi-classification strategy with 20 distinct categories, utilizing a large and pre-annotated dataset. The problem of imbalance was resolved by utilizing the GAN technique. The data is separated into two parts, train, and test, using a two-split. The training set represents 60% of the dataset, which is a relatively small proportion of the training set. The results indicate that BERT with GAN outperformed LMGAN; however, it is notable that there is a lack of comparison with previous studies again.

Moreover, in English studies, *Agarwal et al. (2022)* utilized a BERT-based uncased model to identify sexual predators in conversation files. The researchers employed binary classification on a large dataset that had been pre-annotated. To address the problem of unbalanced data, they resolve it by assigning more weight to the minority class. However, there is a slight issue noticed while dividing the data. They assigned 30% of the dataset for training purposes, while the rest portion was used for testing. This distribution rate is unusual for this study. As a consequence of their comparison with various Transformers based on BERT and with previous studies, the BERT-based uncased model outperformed some of the previous studies and all other models based on BERT.

In addition, *Jamil et al. (2023)* utilized BERT and RoBERTa models to identify dangerous events from both Wikipedia and a text *corpus*. The dataset used was pre-annotated and manually labeled and consisted of multiple classes. The dataset is imbalanced; however, they ensure equal training for each class to achieve balance. And employed three distributions split. In summary, the researchers compared their findings with XLNet but did not compare them with previous studies. They found that BERT performed the best in identifying more tasks, whereas RoBERTa and XLNet performed well in certain tasks.

Further study (*Yu, Li & Feng, 2024*) utilizes BERT and RoBERTa, along with AM-BERT and AM-RoBERTa, to classify documents sourced from newspapers. The researchers utilized a publicly available dataset that was open source, consisting of 20 distinct classes and already annotated. However, the dataset exhibits an imbalance, and they fail to address this constraint. Divided into three distributions. There's no comparison with previous studies. In summary, the findings of AM- RoBERTa, which is a BERT-based model, provide a superior accuracy rate of 90%.

The remaining study contradicts whether BERT-based models enhance classification accuracy, *Bugueno & Mendoza (2020)* classifies English harassment tweets using the BERT base model and compares them using ML and DL models. Their approach involves utilizing binary and multi-category classification techniques on a large dataset. The dataset exhibited an imbalance, which was addressed by employing the SMOTE technique. Before the dataset was balanced, the ML method performed better. However, even after balancing the data, the best results were still not achieved by BERT.

***What are the most common evaluation metrics used in comparison?***
As shown in Table 9, the F1-score and accuracy are the most frequently used metrics. Additionally, precision and recall are also considered, making it appropriate for

**Table 12 High-resource *vs.* low-resource languages.**

| Aspect | High-resource languages | Low-resource languages |
|---|---|---|
| Data availability | Surpasses low-resource languages | Limited availability |
| Accuracy in studies | Higher | Lower |
| Addressing data imbalances | ~80% of studies achieve balance | ~6% of studies address imbalance |

comparison with previous studies. As seen, the high-resource languages were performed almost entirely effectively in BERT-based models when contrasted with the low-resource languages. This is due to the availability of pre-trained models and labeled data.

Furthermore, Table 7 illustrates that 80% of the dataset was either resolved using balance techniques or was already balanced. This contrasts with low-resource languages, which resulted in the majority of the dataset being unbalanced and unresolved. Also, the comparison with previous studies is evident in high-resource studies, which compare nearly all studies, whereas low-resource studies require additional studies to facilitate contrast.

### Is there enough data available in the crime field?

The distribution of studies by language was as follows: 90% were conducted in English, 10% in Chinese. The English and Chinese languages are rich in resources, and most models have been trained using English data.

### High-resource vs. low-resource languages

Table 12 illustrates the contrast between high-resource and low-resource languages for data availability, research accuracy, and the mitigation of data imbalances:

Data availability: High-resource languages are supported by considerably bigger datasets and pre-trained models, whereas low-resource languages have constraints in data availability.

Research in high-resource languages attains greater accuracy owing to the availability of extensive data and resources, in contrast to the diminished accuracy found in some studies in low-resource languages. Despite this, BERT-based models surpass the accuracy of most machine learning and deep learning models in both categories.

Mitigating data imbalances: Around 80% of research in high-resource languages effectively addresses or achieves data balance. Conversely, hardly 6% of studies in low-resource languages endeavor to address imbalances, indicating a substantial disparity in research emphasis and methods.

## Comparison with previous studies

This section will compare and analyze our review to previous SLRs using various criteria, as illustrated in Table 13. The study by *Alammary (2022)* examines Arabic studies within the General domain, the study by *Dharma & Pratama (2023)*, investigates the legal domain without focusing on BERT-based models or the Arabic language, and the study (*Diaz-Garcia & Carvalho, 2025*) examines low and high resource languages, emphasizing high resource languages in the context of illicit activities, particularly with LM and LLM.

**Table 13 Comparison with previous SLR.**

| Criterion | Review article (*Alammary, 2022*) | Review article (*Dharma & Pratama, 2023*) | Review article (*Diaz-Garcia & Carvalho, 2025*) | Our review |
|---|---|---|---|---|
| Title | BERT Models for Arabic text classification: a systematic review | Legal Judgment Prediction: A Systematic Literature Review | A survey of textual cyber abuse detection using cutting-edge language models and large language models | Assessing BERT-based models for Arabic and low-resource languages in crime text classification |
| Journal/Conference | Journal | Conference | Journal | - |
| Domain focus | General NLP Task | Text classification | Text classification | Text classification |
| Languages covered | Arabic | low-resource languages and high-resource languages | low-resource and major in High-resource languages | Arabic, majoring in low-resource languages and high-resource languages |
| Article selection methodology | SLR | SLR | SLR | SLR |
| Number of articles reviewed | 48 | 25 | 68 | 27 |
| Timeframe covered | 2019–2021 | 2018–2023 | 2022–2024 | 2018–2024 |
| Datasets mentioned | Over 10 datasets | 11 private datasets Four public datasets | Over 10 datasets | Over 10 datasets for low-resource languages 10 datasets for high-resource languages |
| Tasks discussed | Sentiment analysis, topic classification | Legal classification | Legal classification | Crime classification |
| Models discussed | Six Arabic BERT models Three multiple support languages BERT model | Over 10 ML and DL models | LMs and LLMs | Seven BERT-based models for low-resource languages Five BERT-based models for high-resource languages |
| Performance metrics | Accuracy, recall, precision, F1-score, AUROC, PRC and PR-AUC | Accuracy, recall, precision, F1-score, and AUC | Accuracy, F1-score | Accuracy, recall, precision, F1-score, sensitivity, specificity, micro F1-scores, WF, macro F1-scores, error, training time, computing power, confusion matrix, F0.5-score, and AUC. |
| Key findings | Arabic BERT models show the best performance over the ML models | ML and DL performed well in the legal field | BERT-based models achieve the best accuracy among the models, especially in the detection of hate speech | Language-specific models are better than multilingual ones. Balancing improves performance, and most studies show BERT models outperform ML and DL models. |

| Criterion | Review article (*Alammary, 2022*) | Review article (*Dharma & Pratama, 2023*) | Review article (*Diaz-Garcia & Carvalho, 2025*) | Our review |
|---|---|---|---|---|
| Future directions | Investigate domain-specific Arabic BERT models | Additional research in the legal field and consideration of the need for a new balancing dataset | Suggested data augmentation for balancing data and emerging LLMs such as GPT and FLAN-T5 shows significant promise, especially in low-resource languages | Construct more balanced labeled data for Arabic and other low-resource languages and apply language BERT-based models that are specifically designed for each. |
| Strengths | Comprehensive model comparison, strong analysis | In-depth focus on the crime field | In-depth focus on the LLMs | In-depth focus on the crime field |
| Limitations | Focus on the general domain | Most of the studies involve traditional ML and DL models, while two studies employ BERT and the legal field's restricted quantity of studies | The scope is broad | The crime field's restricted quantity of studies |

This discrepancy was addressed in our SLR to reflect the specific domain of crime within BERT-based models. As demonstrated in a study by *Dharma & Pratama (2023)*, the scarcity of research is due to the dataset's privacy, as observed in our review. According to the summary provided in the following Table, our review addresses the constraints and examines the strengths and future directions.

## Ethical considerations and challenges

The unavailability of data is compounded by the sensitivity of criminal data since access to law enforcement records or government reports is restricted due to privacy concerns and data confidentiality. These challenges impede the collection of data for further analysis. However, social media platforms offer a paid API for data collection. This facilitates a more comprehensive analysis of the data, but at a high cost, restricts data sharing, and has a limited amount of data accessible.

The challenge lies in the fact that low-resource languages require additional data, even if minimal, or the expense associated with gathering sensitive criminal data necessitates the collection of a modest dataset, which can subsequently be expanded using various data augmentation approaches. Data augmentation refers to the capacity to address the constraints of limited dataset sizes by using synthetic samples to enhance diversity in the training data (*Li, Hou & Che, 2022*).

As an example, the paraphrasing-based method generates new data with minimal semantic variation, the noise-based method introduces discrete or continuous noise while ensuring validity, and the sampling-based method comprehends data distributions to produce novel samples within them (*Li, Hou & Che, 2022*).

The author of the research (*Ghafoor et al., 2021*), examines linguistic variety and asserts that the primary issue is that most non-English languages are resource-limited in machine learning due to the limited availability of labeled datasets. They utilize a single language as a primary medium for comparisons and determine that translating from high-resource to

low-resource languages requires maintaining similar semantics to achieve successful results. Languages such as Arabic, Farsi, Pashto, Bangla, and Urdu can benefit from mutual translation to enhance and increase their labeled datasets.

## Summary of findings

- In the crime domain, approximately 63% of studies that employed BERT-based models were conducted in low-resource languages, while 37% were conducted in high-resource languages.
- Further investigation is necessary in low-resource languages since, in comparison with previous studies, the most effective studies are in high-resource languages.
- English produces the greatest findings, reaching 98%. It is necessary to enhance the outcomes of low-resource languages in comparison to high-resource languages like English.
- The most used models for multilingual crime classification are mBERT, DistilBERT, XLM-R, and XLM-100, subsequently.
- The balancing of data is crucial to mitigate overfitting, bias, or reduced accuracy and can be solved using different techniques as suggested.
- Models developed especially for a language, such as AraBERT in the Study (*El-Alami, Alaoui & Nahnahi, 2022*), and Bangla-BERT in Study (*Sharif & Hoque, 2022*), outperform mBERT, XLM-100, and Distil-BERT in categorizing types of crime.
- A total of 82% of studies do not reference prior studies. For low-resource languages, the field is constrained and necessitates further research; nonetheless, in English, it is crucial to compare with prior studies to identify existing gaps.
- The main evaluation criteria are accuracy, F1-score, precision, and recall. To facilitate comparisons with prior research, it is recommended that the majority implement these metrics.
- The dataset is significantly constrained in low-resource languages, particularly Hausa, Persian, Urdu, and Arabic, since the Study (*El-Alami, Alaoui & Nahnahi, 2022*) employed an API translator to convert text from English to Arabic. Arabic datasets must be established.

## CONCLUSION AND RECOMMENDATION

BERT has seen significant advancements in recent years, particularly in the application of criminal activity classification, demonstrating its effectiveness in identifying illicit behavior. However, most of the research has focused on low-resource languages. This study provides an in-depth analysis of various BERT-based models, assessing their ability to categorize criminal activities in languages beyond English. The goal was to gather and evaluate multiple BERT models studies, compare their performance with traditional ML, DL, and other transformer models, assess the availability of datasets, and determine whether BERT-based models offer improved performance in low-resource languages. The findings of 27 studies indicate that BERT-based models are more effective than ML and DL

models. However, there is a necessity for additional research in the field of crime, particularly in low-resource languages. Additionally, the development of a balanced dataset. Furthermore, the models that were specifically designed to support a specific language are more effective at fine-tuning than the models that support multiple languages.

### Future insight

The review encompassed 27 studies, yielding the following insights: 63% of the studies focused on low-resource languages, Bengali, Arabic, Estonian, Urdu, Indonesian, Hausa, , Pilipino, Persian, Polish, and Hindi. These findings suggest that non-English languages require more extensive research. Several models, including mBERT, XLM-R, DistilBERT, and XLM100, provide multilingual support, making them suitable for languages that lack dedicated models. Furthermore, 82% of the reviewed studies in low-resource languages did not compare their results with previous research, while high-resource languages succeeded in comparing, underscoring the need for more comparative investigations in crime-related applications. The primary evaluation metrics used across studies were accuracy, F1-score, recall, and precision, and future research should consider these metrics for comparison with earlier work.

Data availability remains a significant challenge for low-resource languages, particularly Arabic. Although the scarcity of datasets poses a barrier, the widespread use of social media platforms offers a potential solution. Platforms such as X (formerly Twitter) and Facebook provide APIs for collecting posts, which could facilitate dataset creation. This study identifies critical areas that require further exploration and improvement in the application of BERT-based models for crime classification. The future work will focus on constructing a new Arabic crime dataset and then applying BERT-based models.

### Funding
The authors received no funding for this work.

### Competing Interests
The authors declare that they have no competing interests.

### Author Contributions
- Njood K. Al-harbi conceived and designed the experiments, performed the experiments, analyzed the data, performed the computation work, prepared figures and/or tables, authored or reviewed drafts of the article, and approved the final draft.
- Manal Alghieth conceived and designed the experiments, authored or reviewed drafts of the article, and approved the final draft.

### Data Availability
   This is a literature review.

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
