# Peer review of "Assessing BERT-based models for Arabic and low-resource languages in crime text classification"

_PeerJ Computer Science, doi:10.7717/peerj-cs.3017_

## Round 0.1 · original submission · Major Revisions

I reviewed the paper myself and I tend to agree with the reviewers that the manuscript in the current form is not fit for publication. The manuscript requires a major revision. I will advise the authors to carefully consider all the comments of the reviewers and address them in the revision.

Reviewer 1 ·

Basic reporting

The manuscript makes a good effort to address the underexplored area of applying BERT-based models for crime text classification in low-resource languages. However, while the paper claims to focus on low-resource languages, most of the reviewed studies (approximately 69%) pertain to English, a high-resource language.

This limitation is significant because the challenges and strategies for applying BERT-based models in low-resource languages, such as Arabic and Bengali, differ substantially from those in high-resource languages like English. Without a deeper analysis of low-resource languages, the paper risks overlooking key insights and solutions specific to these contexts.

The paper should:
a) Include a more balanced representation of studies on low-resource languages by broadening the scope of reviewed literature or integrating case studies.
b) Propose or develop methodologies specifically tailored for low-resource language settings, such as data augmentation techniques or unsupervised learning methods for limited datasets.
c) Discuss the distinct challenges low-resource languages face, such as lack of pre-trained models or insufficient annotated datasets, and suggest solutions to mitigate these issues.
d) Elaborate the rationale for focusing on these low-resource languages
e) Provide an in-depth discussion of high-resource versus low-resource languages, particularly regarding how the study addresses these two categories.
f) Critically analyze the gaps in existing research, especially concerning low-resource languages. Most reviewed studies focus on high-resource languages like English, undermining the paper's stated focus on low-resource contexts.
g) address parameter tuning differently for low-resource and high-resource languages in Section 3.6

Experimental design

The paper aligns well with the journal's focus on computational methods and interdisciplinary research. However, while the paper aims to emphasize low-resource languages, the study design is heavily skewed toward high-resource languages. This inconsistency reduces the alignment between the study's objectives and execution.

The use of PRISMA guidelines to review the literature. However, the search strategy is limited to two databases (IEEE Xplore and Web of Science). The author should consider other databases such as Scopus or Google Scholar to get broader coverage of relevant studies, particularly for low-resource languages. The final 16 studies are not sufficient for this topic.

There is limited discussion on the heterogeneity of studies, such as differences in dataset characteristics, model configurations, or crime classifications, which could affect comparability.

Please address the challenges identified in low-resource settings, such as data augmentation methods, etc.

Validity of the findings

Since most of the studies reviewed focus on high-resource languages like English, it undermines the generalizability of findings to low-resource languages.

The answers to the research questions do not consistently distinguish between low-resource and high-resource languages, despite the paper’s stated focus on the former. The findings related to dataset availability are incomplete.
The author should:
a) Provide separate discussions and analyses for low-resource and high-resource languages for each research question to offer a more balanced and comprehensive view.
b) Include more studies or insights on low-resource languages to strengthen the findings and align them with the paper’s focus.

Reviewer 2 ·

Basic reporting

This paper reviewed the model's performance in comparison to Machine Learning, Deep Learning, and other Transformer models in crime text classification.

Experimental design

My comments are shown as follows:
- The authors didn’t explain the architecture of Bert model well.
- They had four research questions, but why they proposed these questions, they didn’t explain it.
- The collected papers are small and they didn’t analyze the collected papers in detail.
- Their experiments are not strong, they gave the results and discussed with a few words.
- They didn’t compare with any research.
- For crime text dataset and classification topics, their review is limitted.
- Technique of the paper is week.
- Their discussion is not sufficient.

Validity of the findings

- I consider what contribution of this paper is.
- They should collect more papers, do more experiments, analyze the result in detail and discuss in dimensions.

Additional comments

The structure of paper is not organized well, they should major revise it

Reviewer 3 ·

Basic reporting

1. Summary of the Article

The article discusses the effectiveness of BERT-based models in text classification within the crime domain, focusing on low-resource languages like Arabic. The authors investigate the performance of BERT-based models when compared with Machine Learning (ML), Deep Learning (DL), and other Transformer models. They explore the availability of datasets and the performance of these models across different languages, presenting a comprehensive analysis supported by scientific articles selected by a review protocol.

2. Strengths
• The study addresses an important gap by focusing on low-resource languages like Arabic.
• Systematic methodology guided by PRISMA enhances reliability.
• Discussion provides clear recommendations for future research.

3. Weaknesses
• The article does not perform new experiments.
• Restricted studies reviewed, limiting the scope of generalizations.

4. Suggestions for Improvement
• Discuss in greater depth the implications of imbalanced datasets for BERT-based models in text classification.
• Discuss techniques to augment datasets.
• Address ethical concerns related to the use of sensitive data.
• Expand the discussion on the scalability of BERT-based models for practical applications.
• Incorporate examples of successful real-world applications of the proposed approaches.
• Conduct additional experiments to assess the effectiveness of BERT-based models using the datasets discussed in the article, focusing on exploring aspects that current state-of-the-art (SOTA) research has yet to address.

5. Recommendation
The article addresses an important issue but requires substantial revisions to improve its depth and scope. I recommend a Major Review to address the limitations highlighted above.

Experimental design

• Conduct a more extensive review of studies to include additional relevant research, e.g.:
Abdalrdha, Zainab Khyioon, Abbas Mohsin Al-Bakry, and Alaa K. Farhan. "Arabic Crime Tweet Filtering and Prediction Using Machine Learning." Iraqi Journal for Computers and Informatics 50.1 (2024): 73-85.
Taiwo, Gbadegesin Adetayo, Muhamad Saraee, and Jimoh Fatai. "Crime Prediction Using Twitter Sentiments and Crime Data." Informatica 48.6 (2024).
Monika, and Aruna Bhat. "Automatic Twitter crime prediction using hybrid wavelet convolutional neural network with world cup optimization." International Journal of Pattern Recognition and Artificial Intelligence 36.05 (2022): 2259005.
Hissah, AL-Saif, and Hmood Al-Dossari. "Detecting and classifying crimes from arabic twitter posts using text mining techniques." International Journal of Advanced Computer Science and Applications 9.10 (2018).
Bahurmuz, Naelah O., et al. "Arabic rumor detection using contextual deep bidirectional language modeling." IEEE Access 10 (2022): 114907-114918.
Abdalrdha, Zainab Khyioon, Abbas Mohsin Al-Bakry, and Alaa K. Farhan. "Improving the CNN Model for Arabic Crime Tweet Detection Based on an Intelligent Dictionary." 2023 16th International Conference on Developments in eSystems Engineering (DeSE). IEEE, 2023.

Validity of the findings

Suggestions for Improvement
• Discuss in greater depth the implications of imbalanced datasets for BERT-based models in text classification.
• Discuss techniques to augment datasets.
• Address ethical concerns related to the use of sensitive data.
• Expand the discussion on the scalability of BERT-based models for practical applications.
• Incorporate examples of successful real-world applications of the proposed approaches.
• Conduct additional experiments to assess the effectiveness of BERT-based models using the datasets discussed in the article, focusing on exploring aspects that current state-of-the-art (SOTA) research has yet to address.

---

## Round 0.2 · Major Revisions

Please carefully consider the comments of Reviewer 2 and revise the manuscript accordingly.

Reviewer 2 ·

Basic reporting

In my comment 3, the collected papers are small and they didn’t analyze the collected papers in detail.
Let us take a look on the revised paper. They have reviewed total 23 papers. for low-resource languages the paper has 13 papers (2020:4, 2021:1, 2022:6, 2023:2), for high-resource languages it has 10 papers (2020:2, 2021:2, 2022:4, 2023:1, 2024:1). This collected paper is small. They should collect more paper and analyze the collected paper in detail.

Experimental design

I keep my comment 4 on the revised paper. Their experiments are not strong. They should do more experiments and analyze the result in detail.
In my comment 5, they didn’t compare with any research. There are many review papers focusing on text classification for Arabic and other low-resource languages, they should compare with these papers.
They discussed 7 Bert-based models and said that these models are most used in the field of crime in Arabic and other low-resource languages, is this right?
By my knowledge, there are many Bert models such as BERT (Base & Large), DistilBERT, ALBERT, TinyBERT, MobileBERT, BioBERT, ClinicalBERT, SciBERT, LegalBERT, FinBERT, mBERT, XLM-R, IndicBERT, CamemBERT, RoBERTa, DeBERTa, ELECTRA, SpanBERT.
For Arabic and other low-resource languages, here are some models: AraBERT (v1, v2,v3,v4), MARBERT, ARBERT, QARiB (Arabic), AfriBERTa (African languages), IndicBERT, MuRIL (Indian languages), ThaiBERT, ViBERT (Southeast Asian languages), mBERT, XLM-R (Multilingual & cross-lingual NLP).
So they should collect more bert-base models and present them in detail.
Actually, crime-related text is not big domain in Arabic and other low-resource languages, they should explain why they only focused on this topic.

Validity of the findings

Lastly, they should discuss in dimensions and give future research on the topic.

Reviewer 3 ·

Basic reporting

The authors have addressed my comments and effectively implemented the suggested improvements. Therefore, I recommend accepting the article.

Experimental design

No comment

Validity of the findings

No comment

---

## Round 0.3 · Major Revisions

A reviewer believes that his/her comments from the previous version are not fully addressed. Please refer to the comments from the current and previous version and address them. As this is the second attempt, please consider this as final revision and address all the shortcomings.

Reviewer 2 ·

Basic reporting

I gave many comments in the previous review, in the second revision, the authors only took comparison with two review papers, only 1 new one in the reference.

Experimental design

I keep my comment 3 in the previous review in this revision: the collected papers are small and they didn’t analyze the collected papers in detail.

Validity of the findings

They should discuss more in the revision.

---

## Round 0.4 · accepted · Accept

The reviewer is satisfied with the revision and is recommending acceptance of the manuscript.

Reviewer 2 ·

Basic reporting

The revision includes my comment

Experimental design

I accept the revision

Validity of the findings

The revision is good

Additional comments

When they reply to my comment they should show where they do that